# ADAPTIVE TOOL USE IN LARGE LANGUAGE MODELS WITH META-COGNITION TRIGGER

## ABSTRACT

Large language models (LLMs) have demonstrated remarkable emergent capabilities, reshaping the landscape of functional tasks by leveraging external tools to tackle complex problems, such as those requiring real-time data or specialized input/output processing. Existing research primarily focuses on equipping LLMs with a broader array of diverse external tools (*e.g.,* program interpreters, search engines, weather/map applications) but overlooks the necessity of tool usage, invoking external tools indiscriminately without assessing their actual need. This naive strategy leads to two significant issues: 1) increased latency due to prolonged processing times, and 2) potential errors arising from communication between LLMs and external tools, resulting in faulty outputs. In this paper, we introduce a concept we term meta-cognition as a proxy for LLM self-capability, and we propose an adaptive decision-making strategy for invoking external tools, referred to as *MeCo*. Specifically, MeCo focuses on representation space to capture emergent representations of high-level cognitive phenomena that quantify the LLM's meta-cognitive scores, thereby guiding decisions on when to use external tools. Notably, MeCo is fine-tuning-free, incurring minimal cost, and our experiments demonstrate that MeCo accurately detects the model's internal cognitive signals. More importantly, our approach significantly enhances decision-making accuracy in tool use for multiple base models across various benchmarks.

## 1 INTRODUCTION

Equipping Large language models (LLMs) with tool learning capabilities represents a promising paradigm to address complex tasks relying on external/real-time sources (Komeili, 2021; Tang et al., 2023), specialized format/schema (Yang et al., 2023; Gao et al., 2023; Lu et al., 2024), domain-specific knowledge (He-Yueya et al., 2023; Schick et al., 2024), and so on. Although existing research has focused on increasing the number and types of tools available within this paradigm (Qin et al., 2023; Hao et al., 2024) and optimizing their usage of these tools (Patil et al., 2023; Shen et al., 2024), the decision-making process regarding when tools are necessary remains underexplored.

The prevailing strategy adopted by existing paradigms, which involves enhancing tool usage by fine-tuning LLMs on carefully crafted datasets, is often hampered by the quality of these datasets. On the other hand, if an LLM always relies on external tools to respond to user queries, it encounters two notable limitations. Primarily, it leads to increased latency (Qu et al., 2024; Wang et al., 2024), as using an external tool (*e.g.,* search engine) can take significantly longer than leveraging the model's internal knowledge. Furthermore, the heavy reliance on external APIs and tools poses risks of robustness and integration issues. It potentially introduces incorrect or inconsistent outputs when tools malfunction (Qin et al., 2023) or are unnecessarily used (Lu et al., 2024; Wu et al., 2024).

To address aforementioned limitations by indiscriminate use of external tools, we propose an adaptive tool use strategy aimed at improving decision-making in LLMs concerning tool utilization. We introduce a novel approach, termed **MeCo**, a **Me**ta-**Co**gnition-oriented trigger that facilitates more judicious use of external tools. We define meta-cognition as the model's ability to self-assess its own capabilities and limitations, discerning whether it can address a user's query independently or if it needs to utilize external tools. Overall, **MeCo** integrates the following key principles:

1. **Meta-Cognition-Oriented Trigger Mechanism:** This core component ensures that LLMs maintain an ongoing assessment of their capabilities and limitations, enabling them to determine the

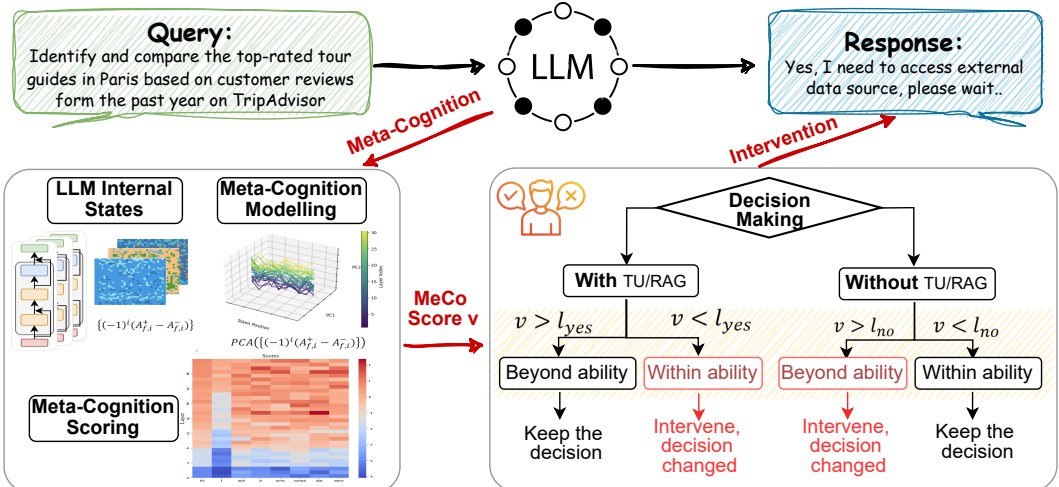

Figure 1: Overview of **MeCo**: Learned Meta-Cognition determines the timing for function calls or retrieval by using a trained meta-cognition probe to detect the internal state of an LLM.

necessity of external tools. This self-awareness is crucial for minimizing unnecessary tool invocation and optimizing the model's performance.

2. **Effective Policy Utilization:** Leveraging the meta-cognition evaluation, we implement a policy that governs tool use based on quantified meta-cognition feedback. Our experimental results demonstrate this policy's superiority over existing methods in decision accuracy.

3. **Generability:** The ability of **MeCo** to generalize across various scenarios is validated empirically, ensuring its effectiveness and robustness in diverse operational environments. Moreover, we treat adaptive Retrieval-Augmented Generation (RAG) as a specific instance of tool utilization and validate the effectiveness of **MeCo** on adaptive RAG against baseline methods.

Building on our initial proposal of an adaptive tool use strategy for LLMs, we detail the development of a computationally efficient plug-in module designed to assess the meta-cognitive states of LLMs. Leveraging the Representation Engineering (RepE) framework (Zou et al., 2023), known for its effectiveness in identifying internal concepts such as honesty and confidence within LLMs, we applied this methodology to detect signals associated with meta-cognition. Our analysis indicates that meta-cognition can generate a strong signal which can be further used to enhance the interpretability of decisions made by LLMs. As illustrated in Figure 1, our strategy dictates that LLMs should engage external tools only when the complexity of a user query surpasses the model's inherent capabilities; otherwise, they should rely on their internal knowledge. Specifically, we establish two thresholds for a given task: one discriminates between strong and weak meta-cognition signals for affirmative responses ("Yes"), and another differentiates these signals for negative responses ("No"). This dual-threshold approach allows us to refine the model's decision-making process, particularly when meta-cognitive signals are weak, suggesting uncertainty or insufficient knowledge.

In summary, our contributions are four-fold: 1) We introduce the concept of adaptive tool use, which enhances both the efficiency and robustness of existing tool learning paradigms in LLMs. 2) We integrate adaptive tool use and adaptive RAG within a unified framework, with their activation driven by a shared strategy based on meta-cognition detection. 3) We provide a benchmark, **MeCa**, to evaluate the effectiveness of our method. 4) We empirically demonstrate that **MeCo** significantly enhances the model's awareness in tool utilization and RAG processes.

## 2 BACKGROUND

Recent research has delved into the internal representations of LLMs to gain insights into their beliefs and interpretability (Bricken et al., 2023; Levinstein & Herrmann, 2024). Studies such as those by Zou et al. (2023) and Liu et al. (2023a) have demonstrated that specific features and signals (*e.g.,* happiness, honesty, and confidence) align with distinct directions within the LLM's representation

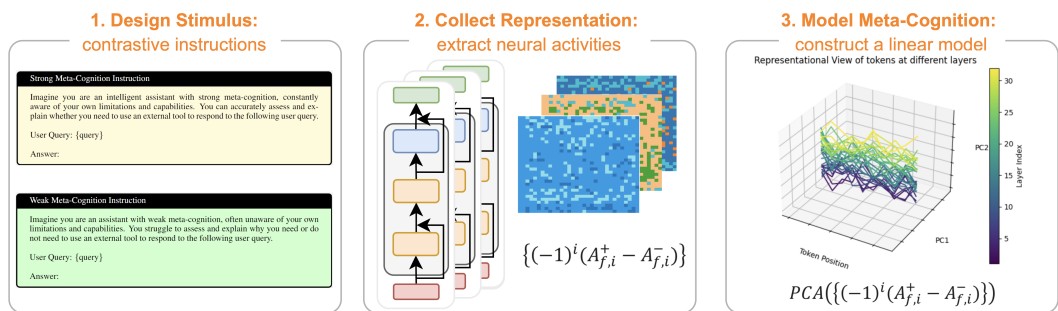

Figure 2: Pipeline for training the meta-cognition probe.

space, making them linearly separable. Figure 2 illustrates the main steps in the pipeline for training various types of probes. To effectively capture and detect these signals, we can utilize contrastive instruction pairs to implicitly induce the emergence of these contrastive signals.

Building on these insights, we enable the capturing and controlling of high-level functions $f$ such as honesty in the model responses. We follow Zou et al. (2023) to design an *experimental prompt* $T_f^+$ that necessitates the execution of the function and a corresponding *reference prompt* $T_f^-$ that does not require the function's execution. An example instruction template might resemble the following:

---

**USER:** ⟨ instruction ⟩ ⟨ experimental/reference prompt ⟩
**ASSISTANT:** ⟨ output ⟩

---

For a function $f$ and a language model $M$, given the instruction response pairs $(q_i, a_i)$ in the set $S$ and denoting a response truncated after token $k$ as $a_i^k$, we collect two sets of internal representations corresponding to the experimental and reference sets:

$$A_f^\pm = \left\{ \text{Rep}(M, T_f^\pm(q_i, a_i^k))[-1] \mid (q_i, a_i) \in S, \text{for } 0 < k \le |a_i| \right\} \tag{1}$$

where Rep represents the representation obtaining operation, $[-1]$ denotes the last token representation of $a^k$, and $A_f^\pm$ are the resulted activations consist of individual vectors.

Our goal is to learn a linear model to identify a direction that predicts the function $A_f^\pm$ based solely on the model's internal representations. Specifically, we apply PCA (Maćkiewicz & Ratajczak, 1993) in an unsupervised manner to pair-wise difference vectors. The first principal component $v_f$, referred to as the reading vector or probe, predicts the function's direction in the model's response. Equation 1 is applied at each layer of $M$ to derive layer-wise probes which are then used to interact with the LLM's representations to monitor and control its behavior.

## 3 APPROACH

In the context of tool use in LLMs, meta-cognition refers to the model's ability to self-assess its own capabilities and limitations to determine whether it can address a user query independently or if it needs to engage external tools. This self-assessment involves evaluating the complexity and requirements of the query in relation to the model's internal knowledge and functions. To quantify meta-cognition, we train a probe that detects the model's level of meta-cognitive awareness. This probe evaluates the rationale behind the model's decision-making process, providing a score that reflects the model's self-assessment accuracy.

For instance, when the model receives a complex mathematical query, the meta-cognition probe evaluates whether it should solve it independently or use an external calculator. A high meta-cognition score indicates that the model accurately recognizes its capabilities or limitations and makes the correct decision, whether to use the tool or not. A low score indicates that the model either incorrectly attempts the task itself or unnecessarily uses the tool.

### 3.1 META-COGNITION PROBE EXTRACTION

The data required to train the meta-cognition probe differs significantly from that used to train the concept such as "honesty" and "confidence" probes. The latter typically involves true-false statements about facts, such as "fire needs oxygen to burn" and "oxygen is harmful to human breathing." These statements are independent of user queries, meaning the model will produce the same statements regardless of the user query.

In contrast, detecting the model's internal cognition regarding whether external tools are needed requires query-dependent responses. To achieve this, we employ leading proprietary LLM to generate user queries related to tool use and their corresponding responses (*i.e.,* Yes/No responses with brief explanations). We then construct the training dataset following the procedures outlined in Section 2. It is important to note that only a small number of queries and responses are enough to train a probe with good performance. The analysis of the relationship between probe performance and the size of the training data is provided in Appendix D. Specifically, after collecting the instruction response pairs $(q_i, a_i)$, where $i$ denotes the index of the queries. We gather the sets of internal representations from the paired data and compute $A_f^{\pm}$ according to Eq. 1, and then apply PCA to the input $\{(-1)^i(A_{f,i}^+ - A_{f,i}^-)\}$ to obtain the first principal component $\nu_f$ as the meta-cognition probe.

After the above training procedures, we will have a probe at each layer in the LLMs (*e.g.,* 32 probes for `Llama-3-8b` models) to detect the meta-cognition signal. We compare the meta-cognition probe with other types of probes that have appeared in existing research, honesty probe Zou et al. (2023) and confidence probe Liu et al. (2024a). We compare the intermediate classification accuracy (in distinguishing between held-out examples where the model is instructed to be honest/confident/have strong meta-cognition or dishonest/unconfident/have weak meta-cognition) of the probes and illustrate the results in Figure 3. Notably, the meta-cognition probe achieves near-optimal accuracy and significantly outperforms its predecessors.

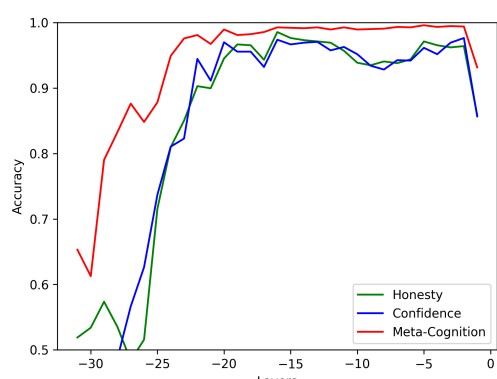

Figure 3: Comparison between different probes. Note that -1 means the last layer in the LLMs.

### 3.2 DECISION-MAKING STRATEGY BASED ON META-COGNITION

After developing accurate probes to detect the model's internal meta-cognition, we design a decision-making strategy utilizing these detection results. Given a user query, the LLM generates a response consisting of $m$ tokens, each associated with a meta-cognition score for every layer in the LLM. This yields a meta-cognition detection array with dimensions $(m, n)$, where $n$ represents the number of layers in the LLM. Our objective is to make a final decision-"Yes", indicating the need to use external tools or RAG, and "No", indicating that LLMs can respond directly without external tools or RAG)-based on this result array.

**Reducing $m$ to 1.** We examine various prompting strategies (detailed in Appendix C) and find that the Yes/No+Explanation strategy, where the model answers with "Yes" or "No" followed by a brief explanation, yields the best performance. Therefore, we focus on the first token of the model's response as it provides a clear signal of whether the model decides to rely on external tools. Extracting the meta-cognition score of the first token to represent the whole response simplifies our decision-making process, as calculating an overall meta-cognition score for the entire response is challenging due to varying response lengths and content across different queries. Since the model always responds with "Yes" or "No" as the first token, basing the trigger mechanism on the first token's meta-cognition score is both reasonable and practical.

**Reducing $n$ to 1.** We select a single probe from the multiple trained probes across different layers to assign the final meta-cognition score to a token. In Zou et al. (2023) and Huang et al. (2023), a mean score from multiple probes' results is usually used to represent the token's final quantification.

However, our experiments show that scores predicted by different probes vary significantly, and simply averaging multiple scores does not yield accurate results. We found that probes in shallower layers (*e.g.,* layer -5 to -2) tend to be more effective, with appropriate score distributions, ranges, and lower variances. Therefore, we use the probe with the highest classification accuracy in the layer -5 to -2 (as shown in Figure 3), as our final probe and rely on its prediction results.

After reducing the meta-cognition results into one scalar value, we adopt the thresholding strategy depicted in Figure 1 to find the optimal thresholds for $l_{yes}$ and $l_{no}$ based on the validation data, which are then applied to the test data.

## 4 BENCHMARK-MECA

**Benchmark Domains:** We evaluate **MeCo** using a public benchmark: Metatool (Huang et al., 2023). In addition, we introduce a new benchmark, named **Me**ta-**C**ognitive Tool **A**ssessment (**MeCa**), where each query underwent a thorough human review. **MeCa** expands on Metatool by incorporating a broader range of scenarios for assessing tool usage. We also include tasks to evaluate adaptive RAG in **MeCa**. Below are the details of these two benchmarks.

**Metatool**: Metatool consists of 1,040 queries designed to assess whether LLMs can recognize when to rely on external tools to solve user queries that they cannot address directly. The tool names and descriptions in Metatool are retrieved from OpenAI's plugin list (OpenAI, 2023), and there are 166 distinct tools in the benchmark. Metatool primarily focuses on evaluating the model's awareness of tool usage for individual tools, where LLMs are provided only with a user query and must independently decide whether or not to resort to external tools without any tool names or descriptions.

Metatool is limited to evaluating user queries without any supplementary information or tool provisions, real-world tasks typically involve more complex intents and a diverse array of requirements. To more accurately reflect these multifaceted scenarios, we developed a new benchmark named **MeCa**. This benchmark is divided into two main components: Tool and RAG. Each query is rigorously reviewed by humans to ensure data quality and relevance. **MeCa** provides several significant improvements over Metatool:

**MeCa-Tool** was meticulously designed to evaluate scenarios where the invocation of external tools is either necessary or unnecessary. **MeCa**-Tool is segmented into the following main categories:

- **Tool Usage Assessment**. This category expands the Metatool to evaluate the decision-making capability of the LLM regarding tool usage in more comprehensive scenarios, specifically whether to invoke any external tools. It includes:
  - Queries that can be handled by the LLM's internal capabilities without external tools.
  - Queries that necessitate the use of one or more external tools, indicating tasks beyond the LLM's standalone capacity.

- **Provided Tool Evaluation**. In this category, the LLM assistant is provided with available tools alongside the user query, with the task of determining tool usage based on the tools' relevance and necessity:
  - Cases where the external tools are unnecessary and the LLM assistant can successfully resolve the queries independently.
  - Situations where external tools are essential and are provided, enabling the LLM assistant to effectively address the user queries.
  - Instances where the required tools to solve the queries are absent from the provided list.

- **Multi-turn Interaction**. This category evaluates the LLM's decision-making regarding tool usage in multi-turn dialogues, involving extended interactions and long context accumulation. This setup tests the LLM's adaptability and decision-making in complex, evolving scenarios that also encompass the aforementioned categories

Based on the main categories outlined, **MeCa**-Tool offers a broader and more varied test set compared to Metatool by integrating diverse scenarios through combinations of these main categories. Specifically, the final configuration covers 6 tasks with 7,000 queries. Please refer to Table 4 in

Appendix A for more details about **MeCa**. To curate the **MeCa**-Tool dataset, we employed a meticulous and structured approach that ensures the queries are relevant to current LLM capabilities. The process unfolded as follows:

1. Collection of diverse scenarios: We began by gathering a broad spectrum of domains and conversational scenarios from various online corpus. This initial step ensures that the subsequent generated synthetic APIs and conversations are grounded in realistic and diverse settings.

2. Synthetic APIs design: Leveraging the collected scenarios, we then synthetically design 500 distinct APIs by emulating examples found in real-world applications, ensuring that they span multiple domains.

3. Query generation: For each query, APIs are randomly sampled from our synthetic APIs pool. User queries are then constructed based on sampled APIs, which may: (i) Require the invocation of the provided APIs; (ii) Not require any tool invocation, relying solely on the LLM's internal knowledge; or (iii) Involve cases where the provided APIs does not include the necessary tools to answer the query directly.

4. Human Verification: After the queries were constructed, they underwent a rigorous human review process. This critical step verified the validity and correctness of the data, ensuring that each query aligns with its intended category and meets quality standards.

Furthermore, as noted in our paper, this work aligns with the emerging trend of adopting an adaptive RAG (Retrieval-Augmented Generation) paradigm, which aims to determine whether a query can be answered directly by the LLM or necessitates external data retrieval. This is because RAG can be viewed as a special case of tool usage, where the LLM's internal knowledge or capabilities are insufficient to address the query, requiring access to external datasets through retrieval tools.

**MeCa-RAG** The "RAG" component was specifically designed to evaluate whether and when retrieval is necessary.

- Positive RAG: cases where the LLM assistant needs to perform retrieval to answer complex queries or queries involving the latest information that LLMs do not have.

- Negative RAG: cases where the LLM assistant can directly respond to simple queries using its internal knowledge, without the need for retrieval.

The dataset was constructed as follows: we selected a subset of fact-based data from the RepE dataset (Zou et al., 2023), which consists of common, well-known facts, such as "The Earth orbits the Sun." These facts were used as model responses, and the leading proprietary LLM (*i.e.,* GPT-4-turbo) was instructed to generate corresponding user queries. Since these queries involve common knowledge that is embedded in LLMs, they do not require retrieval and thus serve as negative RAG examples. For positive RAG examples, we scraped recent news articles from the past few months, ensuring that this content has not been seen by LLMs. We then followed a similar process as mentioned above to generate user queries based on the latest information. This process resulted in queries that require retrieval as they involve knowledge that is unknown or not yet integrated into the LLM's training data. The detailed distribution of **MeCa** can be found in Figure 4.

## 5 EXPERIMENT SETUP

**Baselines:** We evaluate the proposed **MeCo** against two baselines: Naive and $P_{\text{Yes}}$. The Naive baseline determines a "Yes" or "No" based solely on the first token generated by the LLM, where "Yes" represents a positive indication, *i.e.,* requiring external tools, and vice versa. In the $P_{\text{Yes}}$ baseline, we compute a *Yes-score*, as outlined in Equation 2, which offers a more refined measure of the model's confidence compared to the binary Naive approach. Note that the *Yes-score* ranges from 0 to 1, where 0 represents full "No" and 1 denotes full "Yes". The proximity of the *Yes-score* to 0.5 indicates a lower certainty in the model's response, as scores around this midpoint reflect ambiguity in decision-making. $P_{\text{Yes}}$ can adjust the model's output in cases where the *Yes-score* is near 0.5 to enhance the accuracy of both tool use and retrieval timing. Refer to Section C.2 for more details.

$$\text{Yes-score} = \frac{P(\text{Yes} \mid \text{Prompt})}{P(\text{Yes} \mid \text{Prompt}) + P(\text{No} \mid \text{Prompt})} \tag{2}$$

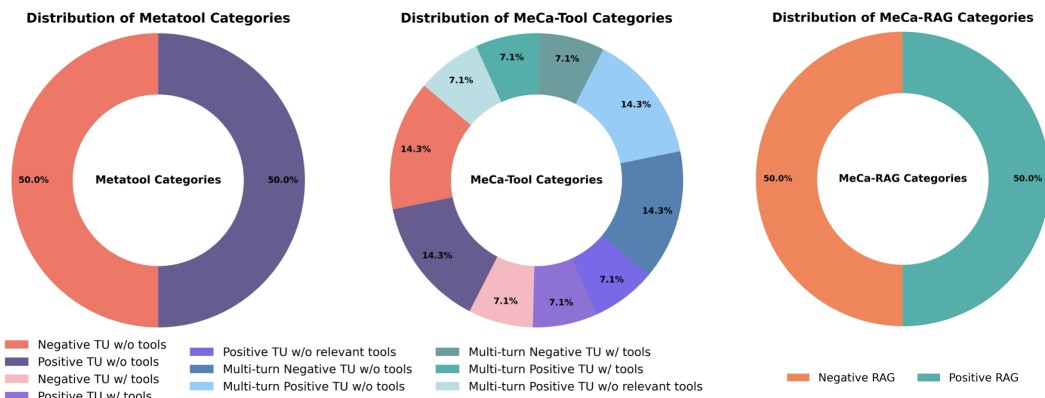

Figure 4: Overview of Benchmarks: Distribution of Metatool, MeCa-Tool, and MeCa-RAG Categories. The Metatool and MeCa-Tool datasets include test data on tool use timing, while the MeCa-RAG dataset focuses on the timing of RAG interactions.

**Backbone LLMs**: We evaluate two widely-used LLMs, *i.e.,* `Llama-3-8b-Instruct` and `Mistral-7b-Instruct-v0.3`. Additionally, to assess the effectiveness of MeCo on larger LLMs, we conducted experiments on `Llama-3-70b-Instruct`. For conciseness, we refer to them as Llama-3-8b, Llama-3-70b, and Mistral-7b throughout the paper, respectively. We also fine-tune these models with data generated by leading proprietary LLM, and the fine-tuned models are denoted as Llama-3-8b-sft, Llama-3-70b-sft, and Mistral-7b-v0.3-sft, respectively.

**Evaluation:** Our experiments primarily focus on the overall accuracy of decisions regarding the necessity of tool use. A tool use decision is considered correct if the query genuinely requires external tools and incorrect otherwise. An analysis of additional performance metrics (including precision, recall, etc.) is provided in Appendix C.

**Prompt:** We explored various prompting strategies, including "Yes/No" responses with or without explanation and the Chain of Thought (CoT; Wei et al. (2022)) approach. Our findings indicate that instructing the model first to provide a "Yes" or "No" response followed by an explanation yields better results than other strategies, including the CoT approach. Detailed results for different prompting strategies are available in Appendix C. Consequently, all experiments in this paper utilize the "Yes/No + Explanation" prompting strategy.

Moreover, we employ two types of prompts in our experiments: 1) prompts with context, which provide specific reasons for why LLMs may require external tools to complete user tasks. These prompts also include five randomly sampled examples to assist the model in making decisions; and 2) prompts without context, which are more concise and contain only the instruction and query. The exact prompts and additional details about the prompt settings can be found in Appendix D.

## 6 EXPERIMENTS

We conduct extensive experiments to empirically reveal the effectiveness of the proposed **MeCo** on two benchmarks: Metatool and **MeCa**. Specifically, we evaluate **MeCo** in adaptive tool use on both Metatool and **MeCa** and in adaptive RAG on **MeCa**.

### 6.1 MECO IN ADAPTIVE TOOL USE

First, we present the distribution of meta-cognition scores collected from the pre-fine-tuning model and post-fine-tuning model in Figure 5. We compare the meta-cognition scores for correct and incorrect responses and visualize how these scores differentiate between correct and incorrect Yes/No answers. Our key observations and interpretations are as follows:

**Clear Gap in Meta-Cognition Scores:** In both pre-fine-tuning and post-fine-tuning experiments, there is a noticeable gap between the meta-cognition scores of correct and incorrect responses. Our

decision-making strategy can identify and leverage this gap to distinguish between correct and incorrect Yes/No answers.

**Higher Scores for Correct Yes, Lower for Correct No**: The meta-cognition scores for correct Yes responses are generally higher than those for incorrect Yes responses. Conversely, the meta-cognition scores for correct No responses are typically lower than those for incorrect No responses. This occurs because the meta-cognition score for Yes/No tokens depends on the token embedding. Therefore, the meta-cognition scores of different tokens are not directly comparable; the score of Yes should only be compared to other Yes scores, and the score of No should only be compared to other No scores.

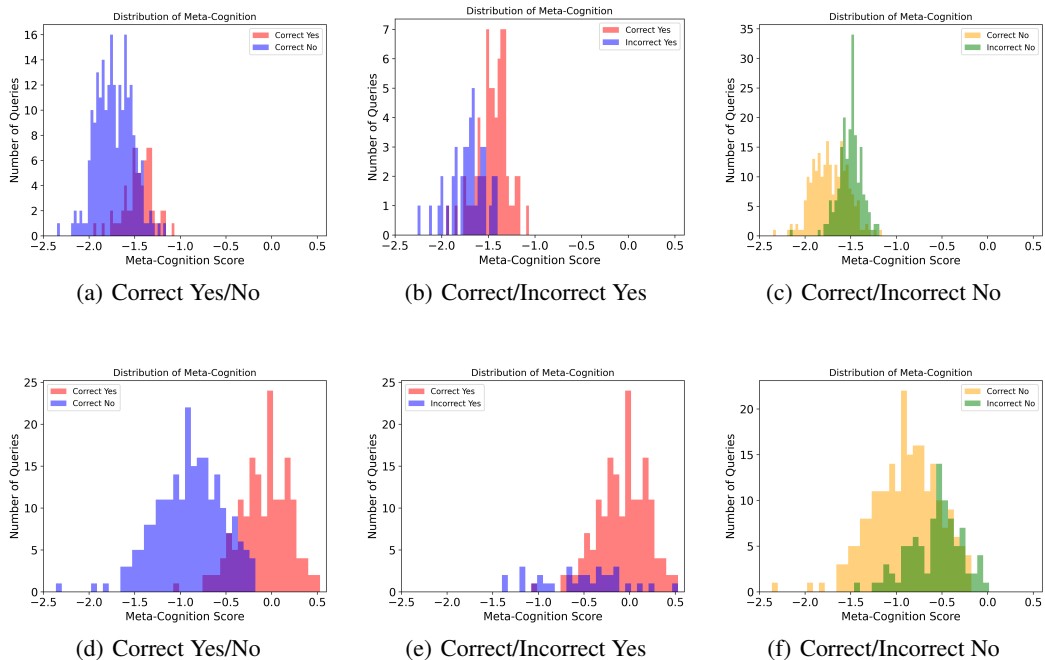

Figure 5: Distribution of meta-cognition scores of the first token in model responses. (a), (b), and (c) are from Llama-3-8b, while (d), (e), and (f) are from Llama-3-8b-sft (post-fine-tuning). The scores are derived from the train data in Metatool, using prompts without context.

Table 1: Performance Comparison between Naive, $P_{\text{Yes}}$ and **MeCo** on Metatool. Note that we are unable to calculate $P_{\text{Yes}}$ or detect the internal states of proprietary LLMs such as GPT-4-turbo.

| LLM | Method | Pre Fine-tuning | | Post Fine-tuning | |
|---|---|---|---|---|---|
| | | With Context | Without Context | With Context | Without Context |
| **Llama-3-8b** | Naive | 61.9% | 58.3% | 82.1% | 80.8% |
| | $P_{\text{Yes}}$ | 63.5% | 62.7% | 81.7% | 80.8% |
| | **MeCo** | **65.0%** | **74.0%** | **84.3%** | **82.3%** |
| **Llama-3-70b** | Naive | 84.6% | 68.8% | 86.0% | 77.7% |
| | $P_{\text{Yes}}$ | 84.8% | 73.7% | 86.2% | 77.1% |
| | **MeCo** | **85.4%** | **79.6%** | **87.3%** | **81.2%** |
| **Mistral-7b** | Naive | 69.0% | 68.5% | 89.2% | 86.0% |
| | $P_{\text{Yes}}$ | 71.2% | 73.1% | 89.2% | 85.0% |
| | **MeCo** | **75.4%** | **74.7%** | **90.2%** | **86.5%** |
| **GPT-4-turbo** | - | 84.4% | 61.3% | - | - |

Table 2: Performance Comparison between Naive, $P_{\text{Yes}}$ and **MeCo** on **MeCa-Tool**.

| Task | Model | Method | Pre Fine-tuning | | Post Fine-tuning | |
|------|-------|--------|-----------------|----------------|------------------|----------------|
| | | | with context | without context | with context | without context |
| Task1 | Llama-3-8b | Naive | 70.0% | 65.0% | 69.0% | 80.0% |
| | | $P_{\text{Yes}}$ | 74.0% | 67.0% | **70.0%** | 78.0% |
| | | **MeCo** | **79.0%** | **72.0%** | 69.0% | **80.0%** |
| | Mistral-7b | Naive | 54.0% | 63.0% | 68.0% | 64.0% |
| | | $P_{\text{Yes}}$ | 54.0% | 63.0% | 69.0% | 63.0% |
| | | **MeCo** | **58.0%** | **67.0%** | **71.0%** | **66.0%** |
| Task2 | Llama-3-8b | Naive | 62.3% | 80.3% | 53.3% | 61.0% |
| | | $P_{\text{Yes}}$ | 78.0% | 80.7% | 58.3% | 68.7% |
| | | **MeCo** | **80.1%** | **81.3%** | **59.9%** | **70.3%** |
| | Mistral-7b | Naive | 42.3% | 55.7% | 52.3% | 53.0% |
| | | $P_{\text{Yes}}$ | 45.0% | 60.0% | 55.3% | 62.3% |
| | | **MeCo** | **66.7%** | **66.0%** | **60.7%** | **66.3%** |
| Task3 | Llama-3-8b | Naive | 54.3% | 78.7% | 59.0% | 68.7% |
| | | $P_{\text{Yes}}$ | 66.0% | **81.3%** | 57.7% | 70.0% |
| | | **MeCo** | **73.3%** | 79.5% | **60.0%** | **73.4%** |
| | Mistral-7b | Naive | 55.7% | 67.3% | 58.3% | 73.7% |
| | | $P_{\text{Yes}}$ | 56.7% | 70.7% | 61.0% | 75.0% |
| | | **MeCo** | **74.8%** | **78.3%** | **65.7%** | **82.0%** |
| Task4 | Llama-3-8b | Naive | 66.0% | 50.0% | 74.0% | 77.0% |
| | | $P_{\text{Yes}}$ | 66.0% | 62.0% | 75.0% | 77.0% |
| | | **MeCo** | **69.0%** | **69.0%** | **75.0%** | **84.5%** |
| | Mistral-7b | Naive | 60.5% | 70.0% | 92.5% | 77.5% |
| | | $P_{\text{Yes}}$ | 66.5% | 71.0% | 92.5% | 80.5% |
| | | **MeCo** | **69.0%** | **78.5%** | **95.0%** | **87.0%** |
| Task5 | Llama-3-8b | Naive | 70.5% | 54.0% | 71.0% | 78.5% |
| | | $P_{\text{Yes}}$ | 72.0% | 71.5% | **80.5%** | **84.0%** |
| | | **MeCo** | **74.0%** | **78.5%** | 79.5% | 82.0% |
| | Mistral-7b | Naive | 73.5% | 76.0% | 87.5% | 82.0% |
| | | $P_{\text{Yes}}$ | 73.0% | 76.0% | 87.5% | **83.0%** |
| | | **MeCo** | **76.2%** | **80.0%** | **88.0%** | 82.0% |
| Task6 | Llama-3-8b | Naive | 60.5% | 53.5% | 78.5% | 83.0% |
| | | $P_{\text{Yes}}$ | 62.0% | 64.5% | **81.5%** | 82.0% |
| | | **MeCo** | **63.5%** | **67.0%** | 80.0% | **86.5%** |
| | Mistral-7b | Naive | 73.0% | 62.5% | 85.0% | 70.5% |
| | | $P_{\text{Yes}}$ | 73.5% | 63.0% | 86.5% | 78.0% |
| | | **MeCo** | **74.0%** | **65.5%** | **88.0%** | **80.5%** |

In our experiment, we sampled a subset of queries from the Metatool benchmark to create training data for determining the optimal thresholds for $P_{\text{Yes}}$ and **MeCo**. We then applied these thresholds to the test queries in both Metatool and **MeCa**-Tool (Task1 and Task4). Due to the fundamental difference between the queries in Metatool and those in Task 2, Task 3, Task 5, and Task 6 in **MeCa**-Tool, we randomly sample 100 queries from each of these categories to serve as hold-out testing data. We fit the thresholds for both $P_{\text{Yes}}$ and **MeCo** using the remaining data. The complete evaluation results are summarized in Table 1 and Table 2. Our key observations are as follows:

**Superiority of MeCo:** On both benchmarks, **MeCo** significantly enhances the model's naive decision accuracy regarding tool use, outperforming $P_{\text{Yes}}$ by a considerable margin, indicating the effectiveness of the meta-cognition-based trigger mechanism. Notably, **MeCo**'s superiority is con-

sistent across multiple backbone models and various evaluation settings, including both with and without context, as well as pre- and post-fine-tuning.

Importantly, the improvement achieved with **MeCo** incurs minimal costs, as it involves a fine-tuning-free and easy-to-integrate module. Note that fine-tuning and MeCo are two orthogonal approaches, and MeCo can provide additional benefits to fine-tuned models. Moreover, fine-tuned models do not transfer well to "out-of-distribution" testing scenarios. For instance, we observed performance degradation in the fine-tuned Llama-3-8b on Task 2 and Task 3 of MeCa-Tool. In contrast, the improvement brought by MeCo is consistent and robust across various testing scenarios.

**MeCo**'s superiority on **MeCa** is particularly promising and significant. **MeCa** contains more complex and realistic queries and user-assistant interactions, closely mimicking real-world scenarios. This underscores MeCo's applicability to real-world LLMs, highlighting its potential for practical deployment and effectiveness in diverse and realistic scenarios.

**Transferability:** The results of Task1 and Task4 in Table 2 indicate that $P_{\text{Yes}}$ and **MeCo**, when fitted on one benchmark, can effectively transfer to other benchmarks. It's worth noting that Metatool and **MeCa** feature different tool sources and styles of queries. We hypothesize that the model's internal cognition is model-dependent, and once fitted on one benchmark, the decision strategy (*i.e.,* the thresholds) can be transferred to other testing datasets. Although it is always better to align the decision strategy with real testing data, **MeCo** demonstrates satisfactory performance even when directly transferred, highlighting its robustness and adaptability.

## 6.2 MeCo in Adaptive RAG

We further evaluate the effectiveness of **MeCo** in the adaptive RAG task, where the LLMs need to determine whether or not to retrieve external information to address the user query. Typically, no reasons or examples are provided to the LLMs in adaptive RAG, and we follow this setting by providing no context in the prompts. The results in Table 3 further validate the effectiveness of **MeCo** in the adaptive RAG task, demonstrating its robustness as a trigger mechanism across various applications. Note that GPT-4-turbo has more up-to-date information and thus does not perform RAG as often as GPT-3.5-turbo and results in a lower accuracy on our benchmark.

Table 3: Performance Comparison between Naive, $P_{\text{Yes}}$ and **MeCo** on **MeCa-RAG**.

| Model | Method | Accuracy |
|---|---|---|
| **Llama-3-8b** | Naive | 63.0% |
| | $P_{\text{Yes}}$ | 75.0% |
| | **MeCo** | **76.0%** |
| **Mistral-7b** | Naive | 84.0% |
| | $P_{\text{Yes}}$ | 84.0% |
| | **MeCo** | **86.0%** |
| **GPT-3.5-turbo** | - | 86.0% |
| **GPT-4-turbo** | - | 84.0% |

## 7 Conclusion

In this paper, we introduce the concept of adaptive tool use to advance existing tool learning paradigms, which typically rely on external tools without discrimination to address user queries. Drawing on insights from representation engineering, we develop a computationally efficient plug-in module, **MeCo**, that assesses the meta-cognitive states of LLMs. Our approach utilizes a meta-cognition probe to detect signals associated with meta-cognition, and leverages these quantification results to inform a decision-making strategy that enables LLMs to make more accurate determinations about when to invoke external tools. To support evaluation, we introduce a new benchmark, **MeCa**, specifically designed to evaluate LLMs' awareness of tool use as well as the timing for retrieval. We empirically validate the effectiveness of **MeCo** using both the Metatool and **MeCa**, demonstrating significant improvements in the model's decision-making accuracy regarding the timing for tool use and retrieval. Our findings suggest that by integrating meta-cognition into the tool usage framework, we can enhance the operational efficiency and decision-making capabilities of LLMs across diverse contexts.

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

## A  MECA STATISTICS

Table 4 summarizes the statistics of the **MeCa**. In Task1 and Task4, the positive queries require a specific external tool to address the user queries, while the negative queries require no external tools and can be solved by the LLM's internal capabilities. In Task2 and Task5, we provide a tool name and its description along with the user query, asking the LLMs to determine whether they need to use this specific tool to address the user queries. The neutral queries in Task2 and Task3 indicate that these queries require external tools, but the provided tool is irrelevant to addressing the user query. In Task3 and Task6, we provide a list of tools (ranging from 2 to 5) along with the user query. For multi-turn queries, there is a dialogue between the user and the LLM assistant, where the assistant needs to determine whether it should rely on external tools to address the user query in the final round of the conversation.

Table 4: Tool Usage Categories and Counts

| Task | Category | Count |
|---|---|---|
| **MeCa**-Tool-Task1 | Positive queries without tools | 500 |
| | Negative queries without tools | 500 |
| **MeCa**-Tool-Task2 | Positive queries with relevant tools | 500 |
| | Negative queries with tools | 500 |
| | Neutral queries with irrelevant tools | 500 |
| **MeCa**-Tool-Task3 | Positive queries with a tool list | 500 |
| | Negative queries with a tool list | 500 |
| | Neutral queries with a tool list | 500 |
| **MeCa**-Tool-Task4 | Multi-turn Negative queries without tools | 500 |
| | Multi-turn Positive queries without tools | 500 |
| **MeCa**-Tool-Task5 | Multi-turn Positive queries with relevant tools | 500 |
| | Multi-turn Negative queries with tools | 500 |
| **MeCa**-Tool-Task6 | Multi-turn Positive queries with a tool list | 500 |
| | Multi-turn Negative queries with a tool list | 500 |
| **MeCa**-RAG | Positive RAG | 150 |
| | Negative RAG | 150 |

We directly transfer the $l_{yes}$ and $l_{no}$ thresholds of MeCo, fitted on the Metatool dataset, to Task1 and Task4 in **MeCa**-Tool, and present the results in Table 2. Because the rest of the tasks in **MeCa**-Tool are very different and more complex than the user queries in MetaTool, we randomly sample 100 queries from each category in Task2, Task3, Task5, and Task6, and use these queries as the hold-out testing data. We use the remaining data to fit the thresholds for $P_{yes}$ and **MeCo**. The complete evaluation results are presented in 2.

## B  RELATED WORK

**Tool Use in LLMs**  LLMs have progressed from understanding and generating human-like text to utilizing external tools based on natural language instructions. This evolution expands their application beyond basic conversational tasks to enable dynamic interactions across diverse functional domains, such as facility management and professional services (Patil et al., 2023; Liu et al., 2023b; Qin et al., 2023; Chen et al., 2023). For example, Toolformer (Schick et al., 2024) enables LLMs to use external tools via simple APIs through a supervised fine-tuning (SFT) model. Liu et al. (2024c) demonstrate strong executable functional API calls across different domains. ToolACE (Liu et al., 2024b) trained on synthesized data, achieves state-of-the-art results on the Berkeley Function-Calling Leaderboard (Yan et al., 2024), even with a relatively small model size of 8B parameters. Despite their growing popularity and capabilities, tool use in LLMs often depends on strategies like verbal feedback, which are hampered by the quality of the datasets used for fine-tuning. Several benchmarks/datasets have been developed to support tool use in a data-centric way, such as API-Bank (Li et al., 2023), which provides a set of tool-use dialogues with various APIs to assess the

LLM's tool use capabilities, Toolalpaca (Tang et al., 2023) constructs a comprehensive tool-use corpus derived from collected real-world APIs, designed specifically to fine-tune LLMs for better tool utilization. ToolBench (Qin et al., 2023) focuses on creating a synthetic instruction-tuning dataset for tool use. However, these methods rely solely on superficial textual information, without probing deeper into the LLM's internal states to explain or justify when and why a tool should be called, resulting in an inability to accurately determine the optimal timing for tool invocation.

**Adaptive RAG**   RAG has shown success in supporting AI systems that require up-to-date information or access domain-specific knowledge, particularly where the scope of queries is not seen in the training data of LLMs (Lewis et al., 2020; Ren et al., 2023; Vu et al., 2023; Izacard et al., 2023). This paper is also consistent with the trend of towards adaptive RAG paradigm, which is designed to assess whether a query can be directly answered by the LLMs or requires external data retrieval (Asai et al., 2023; Jiang et al., 2023). Specifically, a simple query within the LLM's knowledge should be directly answered by the LLMs themselves. On the other hand, for complex queries or questions about data they have not been trained on, RAG intervenes to prevent incorrect out-of-date answers or hallucination (Ji et al., 2023). This mechanism allows RAG to dynamically adjust operational strategies of retrieval-augmented LLMs by assessing the boundary of LLM's self-knowledge and the complexity of the query, thereby minimizing unnecessary computational overhead when the queries are answerable by LLMs themselves. Similar to the LLMs' function-calling, the decision of retrieval timing typically hinges on three primary methods: (i) explicit verbal feedback from LLMs (Ding et al., 2024), (ii) enhancements through fine-tuning (Asai et al., 2023), or (iii) probability-based metrics (Kadavath et al., 2022; Jiang et al., 2023). Specifically, He et al. (2021) proposed enhancing the retrieval time efficiency by computing the probability of the next token via interpolating an LLM with a distribution calculated from the $k$ nearest context-token pairs. Drozdov et al. (2022) further extend $k$NN-LM to the adaptive paradigm by assigning the interpolation coefficient according to the retrieval quality measured by semantic similarity. Asai et al. (2023) introduce Self-RAG to improve generation quality and factuality by enabling adaptive retrieval and self-reflection. In contrast, this paper conceptualizes RAG as an external tool and highlights the importance of understanding the internal states of an LLM when developing the retrieval policy.

**Explainability of LLMs**   However, there is a considerable discrepancy between LLM's decision mechanisms (often based on verbalized responses) and their internal cognition (Zou et al., 2023). The internal workings of LLMs are usually unclear, and this lack of transparency poses unwanted risks in downstream decision-making. Therefore, understanding and interpreting LLMs is crucial for elucidating their behaviors and limitations. To address this challenge, various explanations that provide insights into the inner workings of LLMs have been proposed (Zhao et al., 2024): (i) Probing-based explanations: Probing uses vector representations to measure embedded knowledge (Peters et al., 2018; Jawahar et al., 2019) or examines specific knowledge during the LLM's generation process (Li et al., 2022), (ii) Neuron-level explanation: neuron analysis identifies critical neurons that are essential for model's performance (Antverg & Belinkov, 2021; Bills et al., 2023), (iii) representation engineering (RepE): RepE leverages techniques inspired by cognitive neuroscience to identify and enhance the transparency of LLMs by uncovering their internal cognitive states (Zou et al., 2023). In this paper, we aim to detect the internal cognition of LLMs, and intervene LLM's decisions, *i.e.,* ensuring more precise decisions on tool use and retrieval timing.

## C   EXTENDED RESULTS

### C.1   PROMPTING STRATEGIES

To determine the best prompting strategy for tool use, we explore five prompting strategies with multiple base models. The results are summarized in Table 5.

1. Yes/No + Explanation: The model first answers with "Yes" or "No" and then provides a brief explanation for its decision.

2. Yes/No: The model answers solely with "Yes" or "No," without providing any explanation.

3. No/Yes + Explanation: The model first answers with "No" or "Yes" and then provides a brief explanation for its decision.

4. No/Yes: The model answers solely with "No" or "Yes," without providing any explanation.

5. CoT (Chain of Thought): The model is instructed to think step-by-step, reasoning why it does or does not need external tools to address the user query, and finally concludes its decision with "Yes" or "No."

---

**Chain of Thought Prompting.**

You are an intelligent agent, and you need to constantly be aware of your own limitations. I will provide you with a user's query, and you should assess, based on your own capabilities, whether you need to use external tools to better address the user's query. Typically, there are four reasons why you might need to use external tools:

- A. Solving issues with real-time or external data, databases, or APIs
- B. Handling specialized inputs/outputs
- C. Enhancing domain tasks beyond LLM's capabilities
- D. User customization, personalization, and interaction

Please think step by step, and provide a brief explanation for your decision at first. At last, please conclude with "Yes" if you need to use external tools, or "No" if you do not need external tools.

{Few-shot Examples}

User query: {query}

Answer:

---

Note that there are no results for the CoT prompting strategy for the `Mistral-7b-instruct-v0.3` model. Regardless of the prompts used, the model consistently responds with "Yes/No" at the beginning, followed by an explanation of its decision. This behavior effectively mirrors the Yes/No+Explanation prompting strategy. Based on Table 5, we make the following observations and provide corresponding analysis:

1. Yes/No + Explanation generally performs the best out of the five prompting strategies. This approach provides a clear decision followed by reasoning, enhancing the model's reliability and user trust.

2. CoT is not performing as well as expected. Through close human examination, we found that CoT results in long, complex answers where the model might ultimately conclude with a decision that contrasts with its prior reasoning process. This phenomenon is referred to as reasoning inconsistency, a challenge also reported in the literature(Wei et al., 2022; Lyu et al., 2023). Specifically, LLMs sometimes generate the correct answer following an invalid reasoning path or produce a wrong answer after a correct reasoning process, leading to inconsistency between the derived answer and the reasoning process. In contrast, the "Yes/No-Explanation" prompting strategy does not suffer from this reasoning inconsistency in our experiments, thereby achieving better performance compared to CoT.

3. Yes/No prompting strategy works better than No/Yes prompting. We hypothesize this phenomenon is due to the data format in the pre-training data. For example, there are likely many more Yes/No answers and reasoning processes in the training data compared to No/Yes answers, influencing the model's performance.

We adopt `Llama-3-8b-instruct` and `Mistral-7b-instruct-v0.3` as our backbone models because they exhibit strong performance in adaptive tool use. We exclude `Llama-2-7b-chat` due to its poor performance and lack of discernment regarding the necessity of external tools. Additionally, we exclude `Llama-3.1-8b-instruct` as its performance is almost identical to that of `Llama-3-8b-instruct`.

Table 5: Performance Comparison of Different Prompting Strategies

| Model | Prompting Strategies | Accuracy | Precision | Recall | F1 Score |
|---|---|---|---|---|---|
| Llama-2-7b-chat | Yes/No+Explanation | 0.51 | 0.51 | 1.0 | 0.67 |
| | Yes/No | 0.51 | 0.5 | 1.0 | 0.67 |
| | No/Yes+Explanation | **0.52** | 0.51 | 1.0 | 0.67 |
| | No/Yes | 0.51 | 0.5 | 1.0 | 0.67 |
| | CoT | 0.51 | 0.5 | 0.99 | 0.67 |
| Llama-3-8b-instruct | Yes/No+Explanation | **0.72** | 0.82 | 0.57 | 0.67 |
| | Yes/No | 0.63 | 0.61 | 0.72 | 0.66 |
| | No/Yes+Explanation | 0.52 | 0.51 | 0.99 | 0.67 |
| | No/Yes | 0.5 | 0.5 | 1.0 | 0.67 |
| | CoT | 0.62 | 0.59 | 0.84 | 0.69 |
| Llama-3.1-8b-instruct | Yes/No+Explanation | **0.71** | 0.66 | 0.87 | 0.75 |
| | Yes/No | 0.64 | 0.59 | 0.95 | 0.73 |
| | No/Yes+Explanation | 0.57 | 0.54 | 0.97 | 0.69 |
| | No/Yes | 0.53 | 0.51 | 0.99 | 0.68 |
| | CoT | 0.63 | 0.62 | 0.91 | 0.71 |
| Mistral-7b-instruct-v0.3 | Yes/No+Explanation | **0.74** | 0.68 | 0.89 | 0.77 |
| | Yes/No | 0.70 | 0.64 | 0.92 | 0.75 |
| | No/Yes+Explanation | 0.70 | 0.64 | 0.88 | 0.74 |
| | No/Yes | 0.71 | 0.57 | 0.82 | 0.74 |

## C.2 P(YES) APPROACH

The $P_{\text{Yes}}$ baseline introduces a *Yes-score*, as defined in Equation 2. This score provides a nuanced measure of the model's confidence, refining the binary approach taken by the Naive baseline. The *Yes-score* spans from 0 to 1, where a score of 0 signifies a definite "No" and a score of 1 signifies a definite "Yes". Scores close to 0.5 reflect lower certainty in the model's response, signifying ambiguity in decision-making. By adjusting the model's output in cases where the *Yes-score* is near 0.5 to always "Yes/No" answer, we aim to enhance the accuracy of both tool use and RAG timing. We employ 3 to determine the optimal threshold $l$ for the *Yes-score* based on training data, which is then applied to the test data.

$$\text{Decision} = \begin{cases} \text{Yes} & \text{if } \textit{Yes-score} > l \\ \text{No} & \text{if } \textit{Yes-score} \leq l \end{cases} \tag{3}$$

## C.3 DISTRIBUTION OF P(YES) AND META-COGNITION SCORES

Before delving into the analysis, we provide some background on the concept of calibration in the context of Large Language Models (LLMs). Calibration refers to the alignment between a model's predicted probabilities and the actual likelihood of those predictions being correct. A well-calibrated model generates probability scores that accurately reflect the true probability of its predictions.

In Figure 6, we present the distribution of P(Yes) scores for both correct and incorrect Yes/No decisions. Our key observations are as follows:

1. When the model is given detailed instructions and few-shot examples, it demonstrates poor calibration. As illustrated in Figure 6(a), the distributions of P(Yes) scores for correct and incorrect decisions do not show a clear distinction.

2. Conversely, when the model lacks detailed context and must rely on its internal beliefs to make decisions, it exhibits improved calibration. In Figure 6(b), the peak of the distribution for correct scores clearly deviates from that of incorrect scores.

3. After fine-tuning, the model displays significantly better calibration, as shown in Figures 6(c) and (d). Most correct decisions have P(Yes) scores of either 1 (indicating "Yes") or 0 (indicating "No"), while the P(Yes) scores for incorrect decisions vary between 0 and 1.

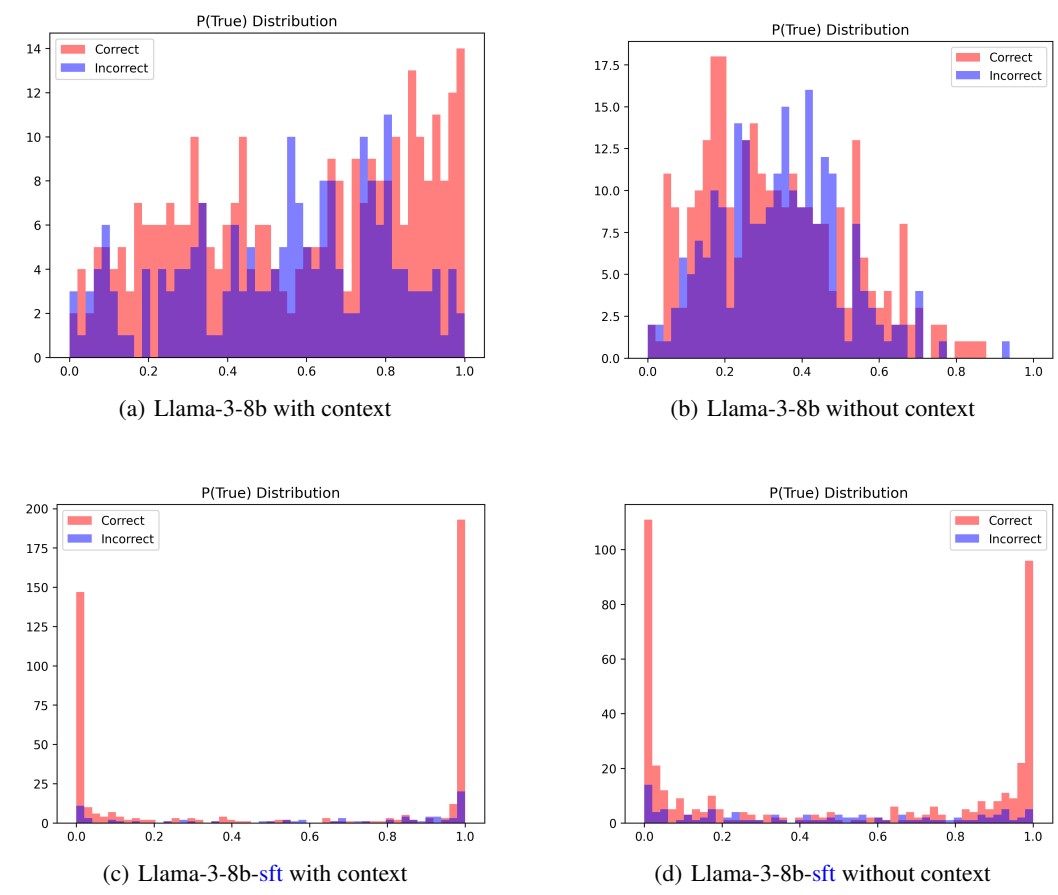

Figure 6: Distribution of the P(Yes) scores of the correct Yes/No and incorrect Yes/No. Llama-3-8b is the model pre-fine-tuning and Llama-3-8b-sft is the model post-fine-tuning. Note that the scores are collected on the training data in the Metatool benchmark.

## C.4 META-COGNITION SCORES AT DIFFERENT LAYERS

We examine the meta-cognition scores at various layers in the model and visualize the results in Figure 7. We focus on the meta-cognition scores at layers -2, -5, -8, -11, and -15 because these layers exhibit the highest classification accuracy, where layer -1 refers to the last layer before the output. Notably, the meta-cognition scores at different layers have distinct value ranges and slightly different distributions. Therefore, it is not reasonable to simply average the scores from different layers as the final score for a token, which has been a common approach in other research works based on RepE. In this study, we use the meta-cognition score from the second-to-last layer as the final score, as this layer demonstrates the highest classification accuracy and effectively differentiates between correct and incorrect responses.

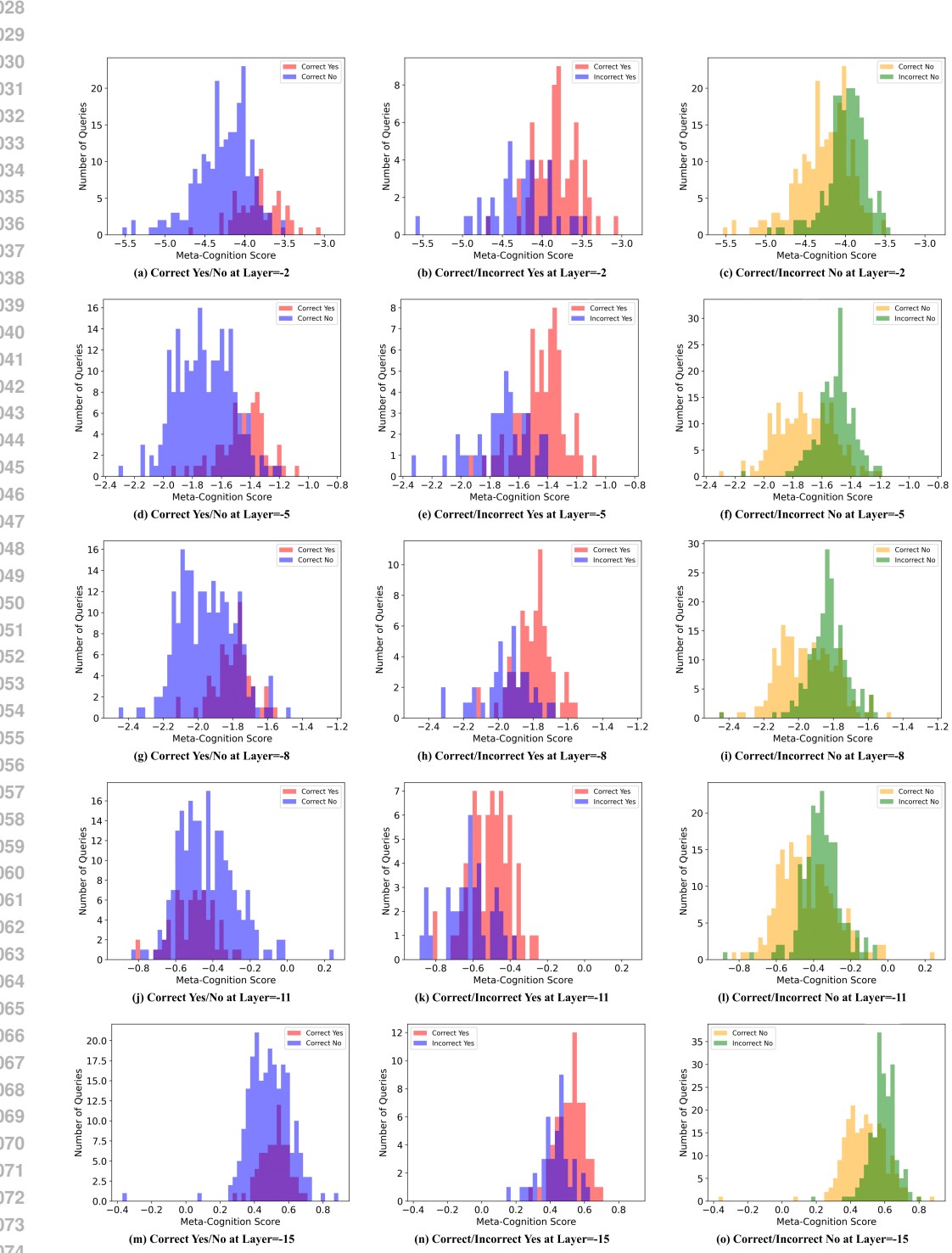

Figure 7: Distribution of meta-cognition scores for the first token at different layers. The results are collected using the Llama-3-8b model on the training data from the Metatool benchmark.

# D    PROBE TRAINING

## D.1    DIFFERENT TRAINING STRATEGIES

Although it increases the length of the instructions and thus may degrade the signal we are detecting, we found that it is much better to provide the model with the query in the instruction than solely instruct the model to follow the ground truth explanations. Therefore, we include the queries in the contrastive instructions below.

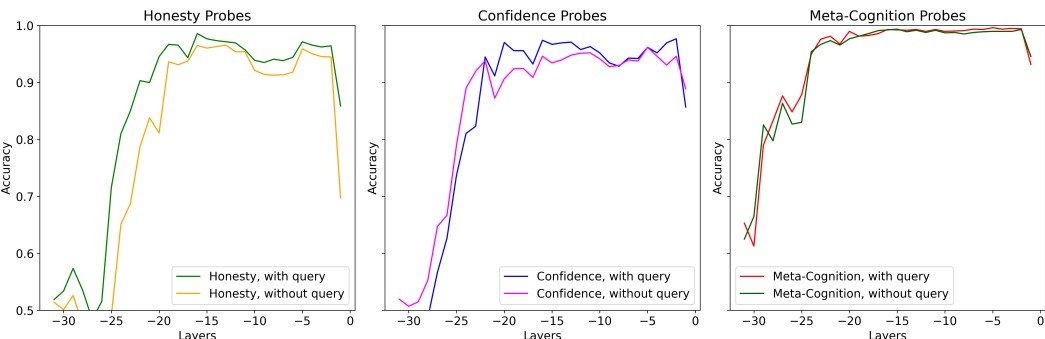

Figure 8: The classification accuracy of different probes trained with the query in the instruction and without the query in the instruction. Training data size is fixed as 2048 in this experiment.

## D.2    DIFFERENT SIZE OF TRAINING DATA

We further examine how the size of the training data affects the outcomes of the meta-cognition probe. Specifically, we analyze the performance of the trained probes with varying sizes of training data, as illustrated in Figure 9 and Figure 10. According to Equation 1, a sentence with 10 tokens can be used to create 10 training data pairs of experimental prompts and reference prompts. Typically, a brief explanation of why or why not to use external tools/RAG corresponds to around 30 to 50 tokens. Thus, a training data size of 256 requires fewer than 10 queries and their associated explanations.

Although different backbone models exhibit significantly varying classification accuracies—with Llama-3-8b achieving the highest and Llama-3-70b the lowest—we found that only a small amount of training data is sufficient to train a probe with near-optimal performance. We hypothesize that this is due to the linear nature of the PCA methods adopted in RepE.

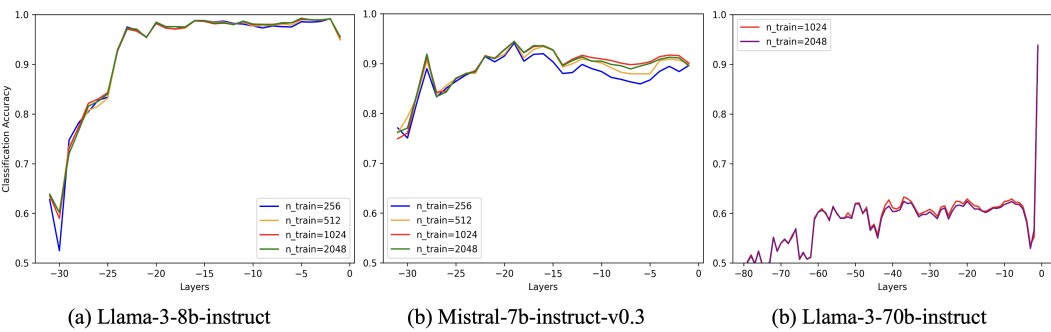

(a) Llama-3-8b-instruct          (b) Mistral-7b-instruct-v0.3          (b) Llama-3-70b-instruct

Figure 9: Training data size vs classification accuracy of meta-cognition probe in adaptive tool use.

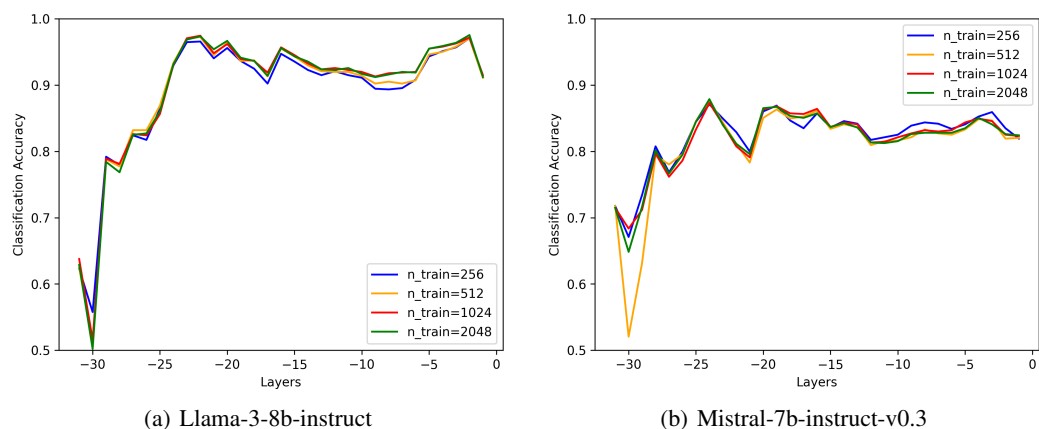

(a) Llama-3-8b-instruct     (b) Mistral-7b-instruct-v0.3

Figure 10: Train data size vs classification accuracy of meta-cognition probe in adaptive RAG.

### D.3 CONTRASTIVE INSTRUCTIONS FOR VARIOUS PROBES

We used the following instruction pair to collect contrastive data and train the honesty probe. Specifically, we instructed the model to be both honest and untruthful when explaining its reasoning for the necessity of tool use.

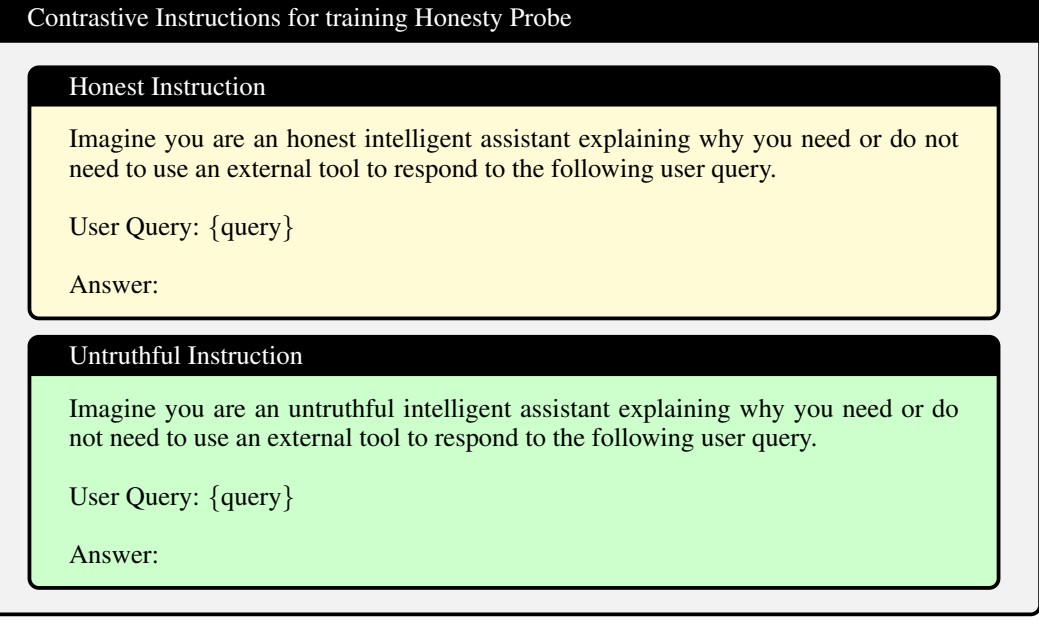

Similarly, we instructed the model to exhibit both confidence and unconfidence when we trained the confidence probe.

---

**Contrastive Instructions for training Confidence Probe**

**Confident Instruction**

Imagine you are a confident intelligent assistant explaining why you need or do not need to use an external tool to respond to the following user query.

User Query: {query}

Answer:

**Unconfident Instruction**

Imagine you are an unconfident intelligent assistant explaining why you need or do not need to use an external tool to respond to the following user query.

User Query: {query}

Answer:

---

For the meta-cognition probe, we instruct the model to exhibit strong meta-cognition by being constantly aware of its own limitations and capabilities and accurately assessing whether an external tool is necessary. Conversely, with weak meta-cognition, the model is often unaware of its own limitations and capabilities and struggles to assess the necessity of tool use.

---

**Contrastive Instructions for training Meta-Cognition Probe**

**Strong Meta-Cognition Instruction in Adaptive Tool Use**

Imagine you are an intelligent assistant with strong meta-cognition, constantly aware of your own limitations and capabilities. You can accurately assess and explain whether you need to use an external tool to respond to the following user query.

User Query: {query}

Answer:

**Weak Meta-Cognition Instruction**

Imagine you are an assistant with weak meta-cognition, often unaware of your own limitations and capabilities. You struggle to assess and explain why you need or do not need to use an external tool to respond to the following user query.

User Query: {query}

Answer:

---

The meta-cognition instruction for Adaptive RAG is similar to that in the adaptive tool use setting, with the only difference being that we replace the necessity of tool use with the necessity of RAG.

> **Contrastive Instructions for training Meta-Cognition Probe in Adaptive RAG**
>
> > **Strong Meta-Cognition Instruction**
> >
> > Imagine you are an intelligent assistant with strong meta-cognition, constantly aware of your own limitations and capabilities. You can accurately assess and explain whether you need to perform Retrieval Augmented Generation (RAG) to respond to the following user query.
> >
> > User Query: {query}
> >
> > Answer:
>
> > **Weak Meta-Cognition Instruction**
> >
> > Imagine you are an assistant with weak meta-cognition, often unaware of your own limitations and capabilities. You struggle to assess and explain why you need or do not need to perform Retrieval Augmented Generation (RAG) to respond to the following user query.
> >
> > User Query: {query}
> >
> > Answer:

# E PROMPTS

## E.1 PROMPTS IN ADAPTIVE TOOL USE

We employ two types of prompts in our experiments: 1) prompts with context, which provide specific reasons for why LLMs may require external tools to complete user tasks. These prompts also include five randomly sampled examples to assist the model in making decisions; and 2) prompts without context, which are more concise and contain only the instruction and query. The exact prompts are provided below. Note that the example queries are randomly sampled in the Metatool benchmark and we follow their setup and don't change the examples associated with queries.

---

**Prompt with context.**

You are an intelligent agent, and you need to constantly be aware of your own limitations. I will provide you with a user's query, and you should assess, based on your own capabilities, whether you need to use external tools to better address the user's query. Typically, there are four reasons why you might need to use external tools:

- A. Solving issues with real-time or external data, databases, or APIs
- B. Handling specialized inputs/outputs
- C. Enhancing domain tasks beyond LLM's capabilities
- D. User customization, personalization, and interaction

If you think it's necessary to use external tools, please respond with "Yes"; otherwise, respond with "No". Additionally, you should provide a very brief explanation for your answer. Here are some examples:

- Query: "Write an opinion piece about why diversity and inclusion is super important for the tech industry. The essay should be targeted at 'tech bros', and should avoid alienating them, but instead appeal to their logic; it should explain how diversity and inclusion of women, immigrants, etc. could benefit them specifically." Answer: No
- Query: "Are there any loopholes that hackers can exploit on my website?" Answer: Yes
- Query: "Plan a weekly lunch menu for a school. Write down a main dish, a carbohydrate side dish, a vegetable side dish, and a dessert for each day." Answer: No
- Query: "Can you break down the main points of this TED talk for me? Here's the YouTube link." Answer: Yes
- Query: "How's the weather in London right now?" Answer: No

User query: {query}

Answer:

---

**Prompt without context.**

You are an intelligent agent, and you need to constantly be aware of your own limitations. I will provide you with a user's query, and you should assess, based on your own capabilities, whether you need to use external tools to better address the user's query. If you think it's necessary to use external tools, please respond with "Yes"; otherwise, respond with "No". Additionally, you should provide a very brief explanation for your answer.

User Query: {query}

Answer:

---

### E.2 PROMPTS IN ADAPTIVE RAG

In adaptive RAG task, LLMs are typically not provided with any reasons or examples to help them make a decision. Following this setting, we conduct the experiments in adaptive RAG without providing context in the prompts as shown below.

> **Prompt without context.**
>
> Imagine you are an intelligent assistant with strong meta-cognition, constantly aware of your own limitations and capabilities. Your task is to accurately assess and explain whether you need to perform Retrieval Augmented Generation (RAG) to respond to the following user query. If you determine that performing RAG is necessary, please respond with "Yes"; otherwise, respond with "No". Additionally, provide a very brief explanation for your decision.
>
> User Query: {query}
>
> Answer:

