# OpenReview forum: "Adaptive Tool Use in Large Language Models with Meta-Cognition Trigger"
_ICLR.cc/2025/Conference — Submitted to ICLR 2025_

### Official Review · Reviewer_h45z · 2024-10-30

**Soundness:** 2
**Presentation:** 2
**Contribution:** 2
**Rating:** 5
**Confidence:** 4

**Summary:**

The authors introduce a method called MeCo, which can enhance the ability of LLM to use external tools.
Unlike existing strategies, the proposed method captures emergent representations within the representation space that quantify the LLM's self-capability, guiding the decision-making process regarding such tools. The method is fine-tuning-free, minimizing additional costs.

**Strengths:**

1. The proposed method is fine-tune-free.

2. A benchmark MeCa is proposed for evaluating the ability of adaptive tool use

**Weaknesses:**

## 1. Meta-cognition is not clear enough and it is easy to confuse readers.
For a self-defined concept, a clear definition and examples should be given. However, the definition in the paper (the sentence below) is vague and lacks clear indicators.

> "We define meta-cognition as the model’s ability to recognize its own capabilities and limitations, discerning whether it can address a user’s query independently or if it needs to utilize external tools."

I don’t know if my understanding is correct, but the meta-cognition in the article is actually self-knowledge. However, the extensive use of meta-cognition in the article confuses this point.
A more detailed definition of Meta-cognition should be put it in a more prominent position. And it would be better if you could provide examples and quantitative indicators.

## 2. The proposed method has limited effectiveness
Table 1 shows that the score improvement brought by fine-tuning is much greater than the method proposed in the paper. Especially on the fine-tuned model, the improvement brought by the proposed method is limited.

Although this method does not require fine-tuning, it does require obtaining the output of the intermediate layer of the model to train the probe. The paper does not compare the overhead of fine-tuning and training probes
In addition, the intermediate layer output is also necessary during inference, which will continue to incur additional overhead.
This results in the actual cost being no lower than fine-tune.

## 3. Insufficient experiments
Only the 7B scale model was tested. The effects on larger models (e.g. 70B) need to be supplemented.
And it is unreasonable to use only the first token of the model to judge correctness, especially for a 7B scale model.

In addition, I have doubts about the prompt in Appendix B.1.
> “Our findings indicate that instructing the model to first provide a “Yes” or “No” response followed by an explanation yields better results than other strategies, including the CoT approach.”

When using the COT method, if still let the model answer “Yes” or “No” first and explain it later, then COT will not produce any effect, because the "Yes" or "No" is not based on the explanation, which is a classic incorrect use of COT. The reason for the results in Table 6 is probably that there is a problem with the prompt itself, but the prompt is not given in the paper. The authors should provide the COT prompt used in the experiment and the method of how to get "Yes" or "No" from LLM's output.

**Questions:**

1. Please give a more detailed definition of Meta-cognition and put it in a more prominent position. It would be best if you could provide examples and quantitative indicators.
2. Please provide experiments on larger models (e.g. 70B level)
3. Please provide the COT prompt used in the experiment and the method of how to get "Yes" or "No" from LLM's output.

---

> ### Author Response · Authors · 2024-11-20
>
> We thank the reviewer for the constructive comments and we have taken the precious advice and made improvements in the revised paper.
>
> **Weakness-1: No clear definition and examples for the self-defined concept.**
> A. See the response in A1.
>
> **Weakness-2: MeCo is outperformed by fine-tuning. Although MeCo is fine-tuning-free, but does not compare the overhead of fine-tuning and MeCo.**
> A. MeCo and fine-tuning are orthogonal approaches, with MeCo capable of being applied to both raw and fine-tuned models, providing additional benefits to the fine-tuning approach. Intuitively, MeCo can be seen as a plugin that enhances adaptive tool use and RAG accuracy by leveraging the model's internal awareness. Importantly, MeCo integrates seamlessly with LLMs, adding negligible latency. In contrast, fine-tuning faces two primary challenges:
>
> 1. Data dependence and scaling effect: The performance of fine-tuned models heavily depends on the availability and quality of the fine-tuning dataset, which is often costly and not always accessible. This requirement can limit the applicability of fine-tuning, especially in environments where such datasets are scarce or incomplete. Besides, the marginal utility of fine-tuning decreases as the size of the fine-tuning data increases, leading to exponentially increasing costs for slightly better performance.
> 2. Limited transferability: Fine-tuned models often underperform in "out-of-distribution" testing scenarios. For instance, in Table 2, the fine-tuned Llama-3-8b showed degraded performance on MeCa-Tool Tasks 2 and 3, which involve "neural" queries that are absent in the fine-tuning data. In contrast, MeCo avoids these limitations, offering consistent performance across diverse scenarios.
>
> We provide the training overheads of both approaches in our experiment. Fine-tuning Llama-3-8b/Mistral-7b required \~4 GPU hours with 4,000 fine-tuning examples on A100 GPUs. In contrast, training the meta-cognition probes and fitting the thresholds for MeCo took \~20 minutes per task. Therefore, MeCo's overhead is significantly lighter than fine-tuning.
>
> MeCo's generality, low overhead, and ability to complement fine-tuning, make it a practical and scalable solution for real-world applications.
>
> **Weakness-3: Insufficient experiments. 1\) small backbone models. 2\) CoT is not clearly explained.**
> A. See the response in A3.
>
> **Q1. Please give a more detailed definition of Meta-cognition and put it in a more prominent position. It would be best if you could provide examples and quantitative indicators.**
> A1. We thank the reviewer for this valuable suggestion. In response, we have included a definition of meta-cognition, along with an illustrative example, at the beginning of Section 3 in the revised paper.  We kindly invite the reviewer to refer to the revised paper for these details.
>
> **Q2. Please provide experiments on larger models (e.g. 70B level)**
> A2. Thank you for the suggestion. We have included Llama-3-70b-Instruct as another backbone model in our experiments. Please refer to Table 1 in the revised paper for the results.
>
> **Q3. Please provide the COT prompt used in the experiment and the method of how to get "Yes" or "No" from LLM's output.**
> A3. Thank you for raising this question. We have added the CoT prompting details in Appendix C.1 of the revised paper. The model is instructed to think step-by-step, reasoning whether it needs external tools to address the user query, and finally concludes with "Yes" or "No." We detect the final "Yes/No" answer by searching for an exact match in the last 150 words of the response. The matching signals for "Yes" and "No" are as follows:
>
> "Yes": 1\) I(i)t is necessary to use; 2\) I(i)t is essential to use; 3\) will need to use; 4\) external tool is necessary; 5\) I need to use the external; 6\) etc.
>
> "No": 1\) I(i)t is not necessary to use; 2\) I(i)t is not essential to use; 3\) will not need to use; 4\) the external tool is not necessary; 5\) do not need to rely on; 6\) etc.
>
> Through close human examination, we found that CoT suffers from reasoning inconsistency, a challenge also reported in the literature\[1,2,3\]. Specifically, LLMs sometimes generate the correct answer following an invalid reasoning path or produce a wrong answer after a correct reasoning process, leading to inconsistency between the derived answer and the reasoning process. In contrast, the "Yes/No-Explanation" prompting strategy does not suffer from this reasoning inconsistency in our experiments, thereby achieving better performance compared to CoT.
>
> \[1\] Wei et al. Chain-of-thought prompting elicits reasoning in large language models. NeurIPS 2022\.
> \[2\] Lyu et al., Faithful Chain-of-Thought Reasoning. The 3rd Conference of the Asia-Pacific Chapter of the Association for Computational Linguistics 2023\.
> \[3\] Zhao et al., A Survey of Large Language Models. arXiv preprint arXiv:2303.18223.

---

> > ### Author Response · Authors · 2024-11-24
> > **Official Comment by authors**
> >
> > Dear Reviewer h45z,
> >
> > In the responses above, we have tried our best to address your questions and clarify any confusion. Given that the rebuttal deadline is approaching, we are eager to engage in a more in-depth discussion with you and welcome any additional suggestions you might have. If you have further suggestions, please let us know, and we will respond as quickly as possible.

---

> > ### Comment · Reviewer_h45z · 2024-11-25
> >
> > Thanks for your kind reply. Most of the concerns have been addressed.
> >
> > The following are some issues that can be discussed, which may not be solved in the short term.
> >
> > 1. Almost all of open source LLMs provide fine-tuned versions, which have stronger tool-use capability and enhanced other capabilities such as multi-round dialogue. Therefore, whether the accuracy can be further improved on the fine-tuned version is a more critical indicator. Or in other words, how to further improve tool use capability while maintaining all the advantages that fine-tune can bring. However, the improvement of the method proposed in this paper is relatively limited.
> >
> > 2. The prompt engineering has a great impact on the behavior of CoT. The CoT prompt provided in the appendix lack the specification of the model output form (e.g., in Markdown or JSON format with reasoning and conclusion tags), which will lead to a certain degree of accuracy decline. Of course, I also admit that the effect of CoT is related to the model capability.
> >
> > In summary, I choose to maintain the current score.

---

> ### Author Response · Authors · 2024-11-25
> **Further Clarifications**
>
> We sincerely appreciate your valuable feedback and constructive engagement throughout the review process. Below, we address your questions in detail and welcome any further concerns you may have.
>
> ---
>
> **Q1. The improvement in fine-tuned models is more critical. The improvement of MeCo is relatively limited.**
> A1. We appreciate the reviewer’s perspective that improving fine-tuned models is crucial for real-world applications. We would like to emphasize that **MeCo consistently enhances the performance of fine-tuned models**, as demonstrated in the tables below. Notably, as a fine-tuning-free, easily integrable, and plug-and-play method, MeCo achieves considerable improvements without introducing substantial latency or training costs. This balance of effectiveness and efficiency highlights MeCo's practical advantages.
>
> Additionally, as widely acknowledged in the research community, the marginal benefits of fine-tuning diminish as the size of the fine-tuning dataset increases. This diminishing return further underscores the value of MeCo: it complements fine-tuned models by delivering additional gains without incurring the high costs typically associated with further fine-tuning.
>
> Thus, MeCo represents a practical and efficient solution for improving tool-use capabilities in both pre-trained and fine-tuned LLMs.
>
> **Table 1. Results on Metatool.**
> | Backbone Model | Method | Post Fine-tuning |  |
> | :---: | ----- | ----- | ----- |
> |  |  | With Context | Without Context |
> | **Llama-3-8b** | Navie | 82.1% | 80.8% |
> |  | MeCo | **84.3% (+2.2%)** | **82.3% (+1.5%)** |
> | **Llama-3-70b** | Navie | 86.0% | 77.7% |
> |  | MeCo | **87.3% (+1.3%)** | **81.2% (+3.5%)** |
> | **Mistral-7b**  | Navie | 89.2% | 86.0% |
> |  | MeCo | **90.2% (+1%)** | **86.5% (+0.5%)** |
>
> **Table 2. Results on MeCa-Tool.**
> | Task | Backbone Model | Method | Post Fine-tuning |  |
> | :---: | :---: | ----- | ----- | ----- |
> |  |  |  | With Context | Without Context |
> | **Task1** | **Llama-3-8b** | Navie | 69.0% | 80.0% |
> |  |  | MeCo | **69.0% (+0%)** | **80.0% (+0%)** |
> |  | **Mistral-7b** | Navie | 68.0% | 64.0% |
> |  |  | MeCo | **71.0% (+3%)** | **66.0% (+2%)** |
> | **Task2** | **Llama-3-8b** | Navie | 53.3% | 61.0% |
> |  |  | MeCo | **59.9% (+6.6%)** | **70.3% (+9.3%)** |
> |  | **Mistral-7b** | Navie | 52.3% | 53.0% |
> |  |  | MeCo | **60.7% (+8.4%)** | **66.3% (+10.3%)** |
> | **Task3** | **Llama-3-8b** | Navie | 59.0% | 68.7% |
> |  |  | MeCo | **60.0% (+1%)** | **73.4% (+4.7%)** |
> |  | **Mistral-7b** | Navie | 58.3% | 73.7% |
> |  |  | MeCo | **65.7% (+7.4%)** | **82.0% (+8.3%)** |
> | **Task4** | **Llama-3-8b** | Navie | 74.0% | 77.0% |
> |  |  | MeCo | **75.0% (+1%)** | **84.5% (+7.5%)** |
> |  | **Mistral-7b** | Navie | 92.5% | 77.5% |
> |  |  | MeCo | **95.0% (+2.5%)** | **87.0% (+9.5%)** |
> | **Task5** | **Llama-3-8b** | Navie | 71.0% | 78.5% |
> |  |  | MeCo | **79.5% (+8.5%)** | **82.0% (+3.5%)** |
> |  | **Mistral-7b** | Navie | 87.5% | 82.0% |
> |  |  | MeCo | **88.0% (+0.5%)** | **82.0% (+0%)** |
> | **Task6** | **Llama-3-8b** | Navie | 78.5% | 83.0% |
> |  |  | MeCo | **80.0% (+1.5%)** | **86.5% (+3.5%)** |
> |  | **Mistral-7b** | Navie | 85.0% | 70.5% |
> |  |  | MeCo | **88.0% (+3%)** | **80.5% (+10%)** |
>
> ---
>
> **Q2. The CoT prompt used in the experiment can be further improved, which will lead to a certain degree of accuracy decline. Of course, I also admit that the effect of CoT is related to the model capability.**
> A2. While we agree that the CoT prompt could be refined, **CoT suffers from two significant limitations** that prevent it from being an ideal solution for adaptive tool use:
>
> 1. **Reasoning inconsistency**: As mentioned in our previous response, CoT often produces inconsistencies between the reasoning process and the final answer \[1, 2, 3\]. Even with carefully designed prompts, CoT has also been observed to negatively affect tool use performance \[4\].
> 2. **High latency and inference costs**: In our experiments, CoT prompting takes significantly more time (3–4 times longer) than the Yes/No+Explanation prompting strategy. This increased latency and computational cost make CoT less suitable for real-world applications where efficiency is paramount.
>
> These drawbacks highlight the practical challenges of CoT, which go beyond prompt optimization.
>
> \[1\] Wei et al. Chain-of-thought prompting elicits reasoning in large language models. NeurIPS 2022\.
> \[2\] Lyu et al., Faithful Chain-of-Thought Reasoning. The 3rd Conference of the Asia-Pacific Chapter of the Association for Computational Linguistics 2023\.
> \[3\] Zhao et al., A Survey of Large Language Models. arXiv preprint arXiv:2303.18223.
> \[4\] ToolACE: Winning the Points of LLM Function Calling. arXiv preprint arXiv:2409.00920.

---

> > ### Author Response · Authors · 2024-11-28
> > **Desire for Re-assessment**
> >
> > Dear Reviewer h45z,
> >
> > As the deadline approaches, we wanted to gently follow up on our recent submission. Your feedback is highly valuable to us, and we would greatly appreciate any updates or further guidance you may have regarding our revisions and responses.
> >
> > We sincerely hope that our responses will encourage a fresh re-evaluation of our work, and we extend our deepest gratitude for your time and consideration.

---

> > ### Comment · Reviewer_h45z · 2024-11-28
> >
> > Sorry for the late reply.
> >
> > For Answer 1, I admit that MeCo has indeed improved the accuracy, but my main concern is that the improvement is not very obvious. I also admit that MeCo itself is very lightweight and has low overhead, but another concern is that MeCo needs to obtain the output of the middle layer of the model when it is used, which may have compatibility issues with existing high-performance inference engines. Unlike the model after fine-tuning, which can be used directly, the modification of the inference engine is also a cost.
> >
> > For Answer 2, there has indeed been work that uses tools with the help of CoT (e.g. [1]). Moreover, the ReAct [2], which is common in practical applications, also uses tools with the help of CoT. CoT does consume longer time and more tokens, but the using of many tools (such as search engines, code executing) is itself time-consuming, which reduces the impact.
> >
> > For the reasons above, I cannot give a rating of 6 or above, so I decided to keep the current score.
> >
> > [1] MultiTool-CoT: GPT-3 Can Use Multiple External Tools with Chain of Thought Prompting. ACL (2) 2023
> >
> > [2] ReAct: Synergizing Reasoning and Acting in Language Models. ICLR 2023

---

> > > ### Author Response · Authors · 2024-11-29
> > >
> > > **W1. For Answer 1, I admit that MeCo has indeed improved the accuracy, but my main concern is that the improvement is not very obvious. I also admit that MeCo itself is very lightweight and has low overhead, but another concern is that MeCo needs to obtain the output of the middle layer of the model when it is used, which may have compatibility issues with existing high-performance inference engines. Unlike the model after fine-tuning, which can be used directly, the modification of the inference engine is also a cost.**
> > > A. We appreciate the reviewer’s thoughtful comments. However, we would like to clarify that MeCo, as a lightweight and low-overhead plug-and-play method, provides significant benefits even on fine-tuned models, as shown in Table 1 of our previous response (with a 1%–4% improvement on SFT models such as Mistral-7B, Llama3-8B, and Llama3-70B). For larger LLMs like Llama3-70B, while fine-tuning shows a modest 1.4% improvement (to 86%), MeCo adds another 1.3% improvement on the SFT model.
> > >
> > > For compatibility with existing high-performance inference engines, MeCo only utilizes the representation of the **first token** ("Yes/No") from **a specific layer** (e.g., layer \-2), as described in Section 3.2. **This minimizes both memory and inference costs, ensuring high compatibility with mainstream libraries.** For example, with Llama3-8B (embedding size 4096\) and FP16 inference, the additional memory cost is only 4096 \* 2 bytes \= 8192 bytes \= 8 KB, which is relatively small. Furthermore, we provide an empirical comparison of inference memory costs with and without MeCo (shown below in Table 3). **Notably, MeCo incurs negligible overhead, less than 0.4% extra memory.**
> > >
> > > MeCo represents a notable improvement over existing approaches \[1, 2\], which rely on representations from all tokens in the response across multiple layers. By focusing only on the first token’s representation, we mitigate the efficiency concerns raised by the reviewer.
> > >
> > > Table 3\. Inference memory cost by different methods.
> > >
> > > | Naive | MeCo |
> > > | ----- | ----- |
> > > | 16.568GB | 16.634GB |
> > >
> > > \[1\] Zou et al., Representation engineering: A top-down approach to AI transparency. arXiv:2310.01405.
> > > \[2\] Liu et al., Ctrla: Adaptive retrieval-augmented generation via probe-guided control. arXiv:2405.18727.
> > >
> > > ---
> > >
> > > **W2. For Answer 2, there has indeed been work that uses tools with the help of CoT (e.g. \[1\]). Moreover, the ReAct \[2\], which is common in practical applications, also uses tools with the help of CoT. CoT does consume longer time and more tokens, but the using of many tools (such as search engines, code executing) is itself time-consuming, which reduces the impact.**
> > >
> > > A. We would like to clarify that the referenced papers (and the point raised by the reviewer) primarily focus on empowering LLM agents to use various types of tools appropriately through prompt engineering. **In contrast, our paper focuses on a different problem – adaptive tool use – which is different and overlooked by the academia, and a critical problem in industry. In fact, MeCo can be seamlessly integrated with CoT/ReAct.** For example, MeCo determines whether or not to use tools at the first step, and Cot/ReAct handles which tools to use and how to use them if needed. MeCo provides additional benefits when combined with other orthogonal methods, such as SFT and CoT/ReAct.
> > >
> > > We respectfully believe this concern is not substantiated, and we kindly request that the reviewer take our response into consideration.

---

> > > > ### Author Response · Authors · 2024-12-02
> > > > **Gratitude for the Pleasant and Constructive Discussion**
> > > >
> > > > As the discussion period nears its conclusion, we would like to once again express our sincere gratitude for your thoughtful reviews and valuable feedback.
> > > >
> > > > If you are satisfied with the revisions (as detailed in the global responses) and the clarifications provided, we would greatly appreciate it if you could consider updating your review to reflect the efforts we have made to address your suggestions and improve the work.
> > > >
> > > > Thank you once again for your time and consideration.

---

> > > > ### Comment · Reviewer_h45z · 2024-12-02
> > > >
> > > > Thank you for your comprehensive response and the additional experimental data.
> > > > I appreciate the efforts made to address my previous concerns. However, these adjustments are not sufficient to convince me to improve the score. Here are some suggestions to make the article better.
> > > >
> > > > 1. Minimize the creation of new concepts:  The manuscript introduces the term "Adaptive tool use" without providing a clear definition, its distinction from conventional tool use, or its significance. This issue was also noted with the term "Meta-cognition" in the initial review.  When introducing new concepts, it is crucial to offer precise definitions and delineate them from established related terms to avoid unnecessary confusion for the readers.
> > > >
> > > > 2. Enhance Methodological Rigor for Improved Accuracy: While the current experimental outcomes demonstrate some level of enhancement, the improvements are relatively modest, particularly concerning larger models. In practical applications, it is these larger models that frequently necessitate the use of tools.

---

### Official Review · Reviewer_FudH · 2024-10-30

**Soundness:** 2
**Presentation:** 2
**Contribution:** 2
**Rating:** 5
**Confidence:** 3

**Summary:**

The paper focuses on the decision making whether an LLM should use an external tool to answer a user query. The authors design a metric to help an LLM recognize its own limits, when answering such queries and if a request is beyond its abilities to decide to use an external tool or RAG. The mechanism for the metric is based on PCA. The authors fine-tune various models to improve the use of that metric with the help of an existing benchmark/dataset (Metatool) as well as two self-created datasets. All three datasets are used to evaluate the resulting models.

**Strengths:**

* proposition of a new metric to help an LLM judge its own capabilities
* two new datasets for judging whether the use of external resources in the forms of tools or RAG is necessary

**Weaknesses:**

The paper lacks focus and flow. The components are only sometimes clearly described and the text contains several contradictions (no fine-tuning according to abstract, but is actually used), for example the decision process shown in the motivation figure is never discussed (and might be wrong) or some discussion seems to lack details like the determination of thresholds. This makes it hard to follow and to clearly grasp, what the contributions are.

* the decision making process presented in Figure 1 is not discussed in the text itself
* section 6: abstracts states that no fine-tuning is used, but apparently fine-tuning is used to improve the generation of the meta-cognition score
* Figure 5: I think the use of different x axis ranges for original model and the fine-tuned model is not great for the comparison of the two.
* size of both self-generated datasets seems kind of small
* number of evaluated models seems kind of small, with the used models having small sizes, which makes me question whether the results would generalize
* table 2: methods does not always perform better (usually one highlights the best performing entries):
  * llama3 with fine-tuning, with context: P_Yes 70%
  * fine-tuned Llama3 model shows not much differences between the three methods
  * llama3 with fine-tuning, without context: Naive 80% (same as MeCo)
* section 6: no discussion on deriving the thresholds for the differentiation
* no ablation studies

additional issues:
* line 142: PCA is never explained nor written out. No reference is provided either.
* Figure 2 is never referenced in the text
* section 4, discussion of MeCa-Tool dataset should use past tense, since it was already assembled, and it does not seem to be a synthetic dataset generator
* section 5, line 332: no reference for CoT
* line 343: "performance references" - apparently those references are missing
  * maybe the whole paragraph was included by accident, since it rephrases the same argument from the previous paragraph
* table 3: placement is not great, since the surrounding text is unrelated
* section 6: "This discrepancy occurs because the meta-cognition score for Yes/No tokens depends not only on the meta-cognition score itself but also on the token embedding."
  * sentence does not make much sense: the score depends on itself as well a token embedding
* prompt examples on pages 19 to 21 should be explained or at least given some context

* minor issues:
  * oversights:
    * line 45: "Lu et al. (2024); Wu et al. (2024)" - remove brackets around the year, and add brackets around the whole citation; should be fixed by using the correct cite command
    * Figure 1, Query: "reviews form the" - it should probably be "from"
    * line 106: "Bricken et al. (2023); Levinstein & Herrmann (2024)" - remove brackets around the year, and add brackets around the whole citation; should be fixed by using the correct cite command
    * line 107: "(Zou et al., 2023; Liu et al., 2023a)" - wrong cite command, here it should actually be "those by Zou et al. (2023) and Liu et al. (2023a) have"
    * line 162: "provided in Section C." - You probably mean "Appendix C".
      * similar for lines 191, 252, 329, 334 and 340
    * lines 234/238: "where LLM assistant" - "an" or "the" is missing before LLM
    * line 253: "To curate MeCa-Tool dataset" - missing "the" before MeCa-Tool
    * Figure 5, caption: "Llama-3-8b-ft" - missing s
    * section 5, Backbone LLMs, line 325: "Llama-3-8b-sft" is referred to as llama-3-sft in Figure 5, please correct the denotation
      * alternatively you could also remove the titles inside the subfigures
    * line 488: "Similar to the LLMs function-calling" - I believe it should LLMs'
    * line 500: "Zou et al. (2023)" - remove brackets around the year, and add brackets around the whole citation; should be fixed by using the correct cite command
    * line 504: "Probing use" - I believe it should be "uses"
    * Figure 6:
      * "Llama-3-8b-ft" should be "Llama-3-8b-sft"
      * additional the title inside the subfigures uses "llama3-8b-inst-sft"
      * "train data" - probably "training data"
    * Figure 7: inconsistent use of "llama3-8b-inst" and "llama3" in the titles of the subfigures
    * Figure 8: "train data" - probably "training data"
    * C.2, title: "train data" - probably "training data"
    * Figure 9: "train data" - probably "training data"
  * references:
    * Bricken et al. 2023: url not clickable
    * Drozdov et al. 2022: cited differently than the other arXiv papers
    * Hao et al. 2024: cited differently than other NeurIPS proceedings
    * He et al. 2021: cited differently than the other arXiv papers
    * He-Yueya et al. 2023: cited differently than the other arXiv papers
    * Huang et al. 2023: cited differently than the other arXiv papers
    * Komeili 2021: cited differently than the other arXiv papers
    * Li et al. 2023: cited differently than the other arXiv papers
    * Liu et al. 2024a/b and 2023a/b/c: cited differently than the other arXiv papers
    * Lu et al. 2024: cited differently than other NeurIPS proceedings
    * Patil et al. 2023: cited differently than the other arXiv papers
    * Qin et al. 2023: cited differently than the other arXiv papers
    * Qu et al. 2024: cited differently than the other arXiv papers
    * Schick et al. 2024: cited differently than other NeurIPS proceedings
    * Shen et al. 2024: cited differently than other NeurIPS proceedings
    * Tang et al. 2023: cited differently than the other arXiv papers
    * Wu et al. 2024: cited differently than the other arXiv papers
    * Yang et al. 2023: cited differently than the other arXiv papers
    * Zou et al. 2023: cited differently than the other arXiv papers

**Questions:**

* Figure 1: Why not use a tool, if the initial decision is to not use a tool, but it also beyond the ability of the LLM?
* section 3.2: The discussion about the probe selection seems rather fuzzy, since detailed results are not shown. How does the selection process generalizes to other models?
* section 5, baselines: Where do P(Yes | Prompt) and P(No | Prompt) come from? Token probabalities of the respective model?
* Will the source code be publicly released?

---

> ### Author Response · Authors · 2024-11-20
>
> We sincerely appreciate the reviewer’s careful and thorough review. We have addressed the raised issues in the revised paper and kindly request the reviewer to consider our responses below:
>
> 1\. Overview Figure: The overview figure is straightforward, and we have discussed the pipeline and logic in the introduction and approach sections. Specifically, the determination of thresholds is described in lines 91, 208, and 381 of the revised paper.
>
> 2\. Fine-Tuning Mention: Fine-tuning is not the core contribution of our paper and is thus not highlighted in the abstract. We implement MeCo on top of fine-tuned models because LLMs are typically fine-tuned to improve performance on downstream tasks, such as adaptive tool use in our paper. This aligns with standard evaluation practices in the literature.
>
> 3\. Figure Visualization: Thank you for the suggestion. We have revised the figure for better visualization.
>
> 4\. MeCa Benchmark Refinement: Post-submission, we continued refining the MeCa benchmark (details provided in Section 4 and Appendix A), increasing its scale, and incorporating more complex and realistic queries to better mimic real-world scenarios. The evaluation results on the updated MeCa benchmark are presented in Table 2 of the revised paper.
>
> 5\. Larger Backbone Model: We have included Llama-3-70b-Instruct as another backbone model and presented the results in the revision. While time constraints prevented us from replicating all experiments on MeCa with Llama-3-70b, we plan to complete this comprehensive evaluation in the next revision.
>
> 6\. 1\) Thank you for identifying the errors; we have fixed them in the revision. 2\) The performance margin decreased after fine-tuning because the model's awareness of tool use significantly improves after fine-tuning, reducing the additional benefit brought by MeCo. However, we want to emphasize that MeCo still provides consistent and notable improvements to the fine-tuned models across two benchmarks and various testing settings. Additionally, MeCo is a fine-tuning-free approach that is easily integrable with any LLMs, offering substantial benefits without incurring high costs.
>
> 7\. Thresholds Derivation: We have discussed the derivation of thresholds in the third paragraph of Section 6.1.
>
> 8\. Ablation Study: We believe that an ablation study is not needed for the evaluation of MeCo.
>
>
> We sincerely appreciate the reviewer’s careful and thorough inspection. We have addressed the issues, oversights, and references in the revised paper and would like to provide clarifications on the following two points:
>
> 1\. **PCA explanation and reference.** We have cited PCA in the revision. PCA is an unsupervised technique primarily used for dimensionality reduction. It identifies the principal components that capture the maximum variance in the data, thereby enabling the learning of principal reading vectors. This method helps in simplifying the data while retaining its most important features.
>
> 2\. **Figure 2 explanation and reference.** We’ve referenced Figure 2 in the revised text. Figure 2 provides a visualization of the pipeline for training the meta-cognition probe and other types of probes. Sections 2 and 3 thoroughly describe the procedures involved in the training process.
>
> **Q1. Figure 1: Why not use a tool, if the initial decision is to not use a tool, but it also beyond the ability of the LLM?**
> A1. Sharp observation; this is indeed a typo in our paper. We have corrected it in the revised version. Please refer to the updated pdf.
>
> **Q2. section 3.2: The discussion about the probe selection seems rather fuzzy, since detailed results are not shown. How does the selection process generalize to other models?**
> A2. The selection process generalizes to all three backbone models used in our paper. While we omitted detailed results in the manuscript for conciseness, we conducted extensive experimentation to systematically evaluate various probes and determine which yielded the best final performance. Our findings consistently showed that probes with higher intermediate classification accuracy contribute to better final decision accuracy. For instance, a probe with an intermediate classification accuracy of 99% typically improves final decision accuracy by 1-2% compared to probes with 95-98% accuracy.
>
> **Q3. section 5, baselines: Where do P(Yes | Prompt) and P(No | Prompt) come from? Token probabilities of the respective model?**
> A3. Correct. We used the logits of the first token in the response from the respective model.
>
> **Q4. Will the source code be publicly released?**
> A4. Yes, we will release the source code soon. We aim to further refine and enrich the MeCa benchmark by incorporating more complex scenarios as suggested by some reviewers before making it publicly available.

---

> > ### Author Response · Authors · 2024-11-24
> > **Official Comment by authors**
> >
> > Dear Reviewer FudH,
> >
> > In the responses above, we have tried our best to address your questions and clarify any confusion. Given that the rebuttal deadline is approaching, we are eager to engage in a more in-depth discussion with you and welcome any additional suggestions you might have. If you have further suggestions, please let us know, and we will respond as quickly as possible.

---

> > > ### Comment · Reviewer_FudH · 2024-11-25
> > >
> > > Thank you for the explanation and the revision.

---

> > > > ### Author Response · Authors · 2024-11-25
> > > > **Official Comment by authors**
> > > >
> > > > Dear Reviewer FudH,
> > > >
> > > > **Thank you for re-evaluating our manuscript and raising the score. We sincerely appreciate your valuable feedback and the constructive engagement throughout the review process.**
> > > > We have carefully addressed your previous suggestions and clarified some misunderstandings. We would be happy to address any further concerns you may have regarding the technical aspects of our work.

---

> > > > > ### Author Response · Authors · 2024-12-02
> > > > > **Individual Response: Gratitude for the Pleasant and Constructive Discussion**
> > > > >
> > > > > As the discussion period nears its conclusion, we would like to once again express our sincere gratitude for your thoughtful reviews and valuable feedback.
> > > > >
> > > > > If you are satisfied with the revisions (as detailed in the global responses) and the clarifications provided, we would greatly appreciate it if you could consider updating your review to reflect the efforts we have made to address your suggestions and improve the work.
> > > > >
> > > > > Thank you once again for your time and consideration.

---

### Official Review · Reviewer_QSLD · 2024-10-31

**Soundness:** 3
**Presentation:** 3
**Contribution:** 3
**Rating:** 6
**Confidence:** 3

**Summary:**

This paper introduces "MeCo," a meta-cognitive mechanism designed for large language models (LLMs) to adaptively determine when to invoke external tools or rely on internal knowledge. The framework centers on "meta-cognition" to gauge LLMs' self-assessed capability to handle queries, with the goal of minimizing unnecessary tool usage, which may increase latency and errors. Through the representation of high-level cognitive phenomena, MeCo detects internal cognitive signals without fine-tuning, using a meta-cognition probe trained on contrastive instructions to determine when tool engagement is needed. Experimentally, MeCo improves decision-making accuracy in adaptive tool use and retrieval-augmented generation tasks, surpassing baseline approaches.

**Strengths:**

Meta-Cognition Trigger Mechanism: The paper introduces a meta-cognition-oriented trigger mechanism for large language models (LLMs), which enables models to assess their own capabilities and invoke external tools only when needed. This approach optimizes efficiency by minimizing unnecessary tool usage​.

Policy Utilization Effectiveness: By integrating meta-cognition evaluations into decision-making policies, the approach improves decision accuracy, proving more effective than prior methods in guiding when and how tools are engaged​.

Generability: The model demonstrates strong empirical adaptability across varied scenarios, confirming the robustness and wide applicability of its meta-cognitive strategy in different environments​.

Benchmark Introduction: The paper establishes a new benchmark, MeCa, to evaluate meta-cognitive strategies in LLMs, setting a valuable standard for future research in adaptive tool use and Retrieval-Augmented Generation (RAG) processes​

**Weaknesses:**

Simplified Benchmarks: The paper primarily evaluates its approach on benchmarks that may not fully reflect real-world complexity. This can limit the broader applicability and relevance of its findings in practical scenarios​.

Underexplored Limitations of Meta-Cognition Scoring: While the meta-cognition approach is promising, the paper does not deeply address cases where this scoring might fail or where it could lead to suboptimal decisions, particularly with ambiguous or highly nuanced queries.

Lack of Robust Comparative Analysis: The analysis lacks a detailed comparison against alternative adaptive approaches. Without this, it’s challenging to assess how the proposed model's efficiency and accuracy improvements stand relative to other recent innovations in adaptive retrieval or tool use​.

Scalability Concerns in Diverse Operational Environments: The paper suggests that the model generalizes well but does not provide sufficient evidence to validate this across varied and complex environments, where scalability might be affected​.

**Questions:**

How does MeCo handle real-world scenarios with complex or ambiguous questions? Were any experiments conducted in more open-ended, unstructured environments, and if so, what were the results?

Can the meta-cognition scoring approach manage ambiguous or nuanced queries that may require partial or iterative tool engagement? If not, how does the system handle such edge cases?

Have the authors tested MeCo’s scalability in more diverse and high-stakes environments where model latency or tool usage frequency could impact outcomes significantly?

How does MeCo compare to other adaptive approaches like reinforcement learning-based or rule-based systems? Were any such methods considered for direct comparison, especially for efficiency or accuracy?

Could the authors clarify any limitations they see in the MeCa benchmark? Are there aspects of meta-cognitive performance that the benchmark doesn’t capture, and are there plans to address them?

---

> ### Author Response · Authors · 2024-11-20
>
> We thank the reviewer for the positive support and constructive comments.
>
> **Weakness-1. Simplified benchmark.**
> A. Our updated results in Table 2 of the revised paper should address this concern. Post-submission, we continued refining the MeCa benchmark (details provided in Section 4 and Appendix A), increasing its scale and incorporating more complex and realistic queries to better mimic real-world scenarios. MeCa now includes 7000 new queries specifically designed to assess adaptive tool usage across six tasks and incorporates various scenarios and interaction lengths, such as multi-turn dialogues.
>
> **Weakness-2. Underexploration of meta-cognition, where it could fail and lead to sub-optimal decisions.**
> A. As shown in Figure 5, a clear gap between the meta-cognition scores of correct and incorrect responses allows our decision-making strategy to effectively differentiate correct from incorrect Yes/No decisions. While we acknowledge that MeCo may occasionally misclassify a correct response due to inaccurate meta-cognition feedback, particularly for ambiguous queries, this potential error is outweighed by the significant benefits of MeCo, as demonstrated in our empirical results.
>
> **Weakness-3. Lack of comparison to other adaptive algorithms.**
> A. See the response in A4.
>
> **Weakness-4. Scalability concerns and lack of experiments in real-world queries?**
> A. We have conducted experiments with the Llama-3-70b model and validated the effectiveness of MeCo on the updated MeCa benchmark, which incorporates more complex and realistic queries. Please refer to the global response and the revised paper for the updated results.
>
> **Q1. How does MeCo handle real-world scenarios with complex or ambiguous questions? Were any experiments conducted in more open-ended, unstructured environments, and if so, what were the results?**
> A1. We have refined the MeCa benchmark to incorporate more complex and realistic queries (over 7000 queries). Please refer to the global response and the revised paper for the updated results.
>
> **Q2. Can the meta-cognition scoring approach manage ambiguous queries that may require partial or iterative tool engagement? If not, how does the system handle such edge cases?**
> A2. Partial or iterative tool use scenarios are effectively managed within our evaluation framework, as they essentially involve sequences or combinations of single tool use decisions. All queries are processed using the decision-making strategy illustrated in Figure 1\. In our experiments, we treat all queries equally based on their meta-cognition scores without applying specific decision rules for ambiguous cases.
>
> We acknowledge that boundary cases may require more sophisticated handling. For such edge cases, a hybrid decision strategy (e.g., combining MeCo with rule-based methods) might improve the user experience. While the development of such hybrid strategies is beyond the scope of this paper, it represents a promising direction for future research. Our contributions, particularly the adaptive tool use framework and meta-cognition system, provide a strong foundation for future advancements in this area.
>
> **Q3. Have the authors tested MeCo’s scalability in more diverse and high-stakes environments where model latency or tool usage frequency could impact outcomes significantly?**
> A3:  Currently, no established benchmarks specifically target high-stakes environments. MeCa represents the most comprehensive benchmark currently available. While we recognize the importance of such high-stakes environments, the development and standardization of relevant benchmarks are ongoing. Future work will aim to assess MeCo in these more demanding settings as appropriate benchmarks become available.
>
> **Q4. How does MeCo compare to other adaptive approaches like reinforcement learning-based or rule-based systems? Were any such methods considered for direct comparison, especially for efficiency or accuracy?**
> A4. Existing frameworks in the literature indiscriminately rely on external tools without adaptiveness. To our knowledge, there are no reinforcement learning-based or rule-based approaches designed for adaptive tool use in this context. Therefore, such comparisons were not included in our experiments
>
> **Q5. Could the authors clarify any limitations they see in the MeCa benchmark? Are there aspects of meta-cognitive performance that the benchmark doesn’t capture, and are there plans to address them?**
> A5. Post-submission, we have improved the MeCa benchmark and will keep refining it with more complex and realistic scenarios. A few potential tasks are:
>
> 1. Tool Costs and Risks: Incorporating varying costs and risks associated with different tools, as highlighted by Reviewer-AwNJ.
> 2. Multi-round and Multi-tool Interactions: Adding scenarios with multi-round interactions with dependencies between tool invocations. This will significantly increase the benchmark’s complexity and relevance to real-world scenarios.

---

> > ### Author Response · Authors · 2024-11-24
> > **Official Comment by authors**
> >
> > Dear Reviewer QSLD
> >
> > As the deadline is nearing, we wanted to gently follow up on our recent submission. Your feedback is highly valuable to us, and we would appreciate any updates or further guidance you might have regarding our revisions and responses.
> >
> > Thank you for your time and consideration.

---

> > > ### Author Response · Authors · 2024-11-28
> > > **Desire for Re-assessment**
> > >
> > > Dear Reviewer QSLD,
> > >
> > > As the deadline approaches, we wanted to gently follow up on our recent submission. Your feedback is highly valuable to us, and we would greatly appreciate any updates or further guidance you may have regarding our revisions and responses.
> > >
> > > We sincerely hope that our responses will encourage a fresh re-evaluation of our work, and we extend our deepest gratitude for your time and consideration.

---

> > > > ### Author Response · Authors · 2024-12-02
> > > > **Gratitude for the Pleasant and Constructive Discussion**
> > > >
> > > > As the discussion period nears its conclusion, we would like to once again express our sincere gratitude for your thoughtful reviews and valuable feedback.
> > > >
> > > > If you are satisfied with the revisions (as detailed in the global responses) and the clarifications provided, we would greatly appreciate it if you could consider updating your review to reflect the efforts we have made to address your suggestions and improve the work.
> > > >
> > > > Thank you once again for your time and consideration.

---

### Official Review · Reviewer_AwNJ · 2024-11-03

**Soundness:** 3
**Presentation:** 3
**Contribution:** 2
**Rating:** 5
**Confidence:** 3

**Summary:**

The paper addresses a practical problem in LLM tool use - when should models actually call external tools versus use internal knowledge. Current approaches tend to call tools indiscriminately, leading to increased latency and errors. The authors propose MeCo, which uses representation engineering (RepE) to detect "meta-cognition" signals that indicate whether a model needs external tools.

**Strengths:**

The problem identification and motivation are excellent. The authors clearly articulate why indiscriminate tool use is problematic and provide compelling examples. The empirical results are strong, showing an 11% improvement in accuracy across various benchmarks. The approach is also practical - it requires no fine-tuning and can be easily integrated into existing systems.

**Weaknesses:**

I am unsure of any new technical details beyond just applying an existing RePe research to tool use. While the authors frame this as detecting "meta-cognition", it's functionally very similar to previous work on detecting other concepts like honesty or confidence. The main innovation seems to be in the framing rather than the technical approach.

The decision mechanism is overly simplistic, using basic thresholds on the meta-cognition scores without any principled way to set these thresholds. There's no consideration of different tools having different costs or risks, or of the model's confidence in its decisions--which seems quite relevant here.

**Questions:**

Que: How does detecting "meta-cognition" differ technically from detecting these other concepts? The paper shows strong empirical results - is this because meta-cognition is particularly well-suited to RepE detection compared to other concepts? Are there any such insights in the results?

---

> ### Author Response · Authors · 2024-11-20
>
> We appreciate the reviewer's insight and will work towards integrating these considerations in future iterations of our research.
>
> **Weakness-1. Technical Contribution is low.**
> A. Our contributions are substantial and extend beyond mere technical contributions.
>
> First, we introduced the adaptive tool use framework, a concept that has been largely overlooked in both academic and industry communities. This framework addresses a critical gap by focusing on the strategic timing of tool usage, a crucial aspect in real-world applications. Moreover, we are the first to integrate adaptive tool use and adaptive RAG within a unified framework, and provide a unified strategy based on our proposed MeCo approach.
>
> Second, we empirically demonstrate that MeCo significantly enhances the model's awareness in adaptive tool use and RAG with minimal training required. Meanwhile, our analysis shows that MeCo provides stronger signals and higher accuracy, capturing a model's internal cognition more effectively than existing metrics like honesty and confidence. Moreover, we advanced the methodology for utilizing detection scores and trained probes (as described in Section 3.2).
>
> Third, we introduced a new benchmark, MeCa, which greatly expands the existing Metatool benchmark (which contains 1040 queries). MeCa includes 7,000 new queries specifically designed to assess tool usage across six main tasks and incorporates various scenarios and interaction lengths, such as multi-turn dialogues. Moreover, we added 300 queries related to adaptive RAG, allowing MeCa to evaluate the effectiveness of both adaptive tool usage and RAG within a unified framework.
>
> **Weakness-2. The decision mechanism is simplistic and not principled. Different tools are viewed equally.**
> A. We have validated the effectiveness of the simple decision mechanism employed by MeCo through extensive empirical results across multiple benchmarks and various evaluation settings. Additionally, we conducted experiments with three backbone models: Llama-3-8b, Llama-3-70b, and Mistral-7b, demonstrating the generality of MeCo across various LLMs. Instead of being a shortcoming, we believe the simplicity and generality of MeCo should be seen as an advantage, allowing for easy and fast integration with any LLMs.
>
> The notion of different tools having different costs or risks is indeed interesting, and we will incorporate this perspective in further developing the MeCa benchmark. As the pioneering paper introducing the concept of adaptive tool use, we initially treated various tools equally in the evaluation to establish a baseline. It is worth noting that this approach is consistent with most existing literature on LLM tool use, where tools are generally treated equally. We believe that addressing tool differentiation is a valuable direction for future work, to be considered once the adaptive tool use framework has gained wider acceptance.
>
> **Q1. How does detecting "meta-cognition" differ technically from detecting these other concepts? The paper shows strong empirical results \- is this because meta-cognition is particularly well-suited to RepE compared to other concepts? Are there any such insights in the results?**
> A1. Detecting "meta-cognition" differs from detecting honesty or confidence in two key aspects:
>
> 1. Training data: Unlike honesty and confidence probes, which rely on factual statements independent of user queries, meta-cognition requires task-specific data that reflects the model’s self-awareness of its capability to address queries. This is especially important for tasks involving external tool use. Our training data consists of user queries related to tool use, accompanied by the corresponding decisions and explanations. Detailed data construction and contrastive instructions are provided in Section 3.1 and Appendix D.3.
> 2. Methodology: We have refined the methodology for utilizing meta-cognition scores. Rather than using the common mean-pooling approach, we focus on scores with the highest intermediate classification accuracy (as described in Section 3.2, "Reducing n to 1"). This provides a more accurate reflection of the model's self-awareness, improving the precision of meta-cognition score utilization and further enhancing the effectiveness of our method.
>
> Our empirical results show that meta-cognition consistently outperforms concepts like honesty and confidence in detecting a model’s internal awareness of its capabilities and limitations. As shown in Figure 3, meta-cognition scores yield higher intermediate classification accuracy, indicating meta-cognition more effectively captures the cognitive processes for decisions on adaptive tool use and RAG. Additionally, Figure 5 highlights a clear gap between the meta-cognition scores for correct and incorrect responses,  which our decision-making strategy exploits to discern correct from incorrect Yes/No decisions. For further analysis on why meta-cognition outperforms, see Appendices C.3 and C.4.

---

> ### Author Response · Authors · 2024-11-24
> **Official Comment by authors**
>
> Dear Reviewer AwNJ,
>
> In the above responses, we have tried our best to answer your questions and resolve your confusion. As the rebuttal deadline is approaching, we are very willing to have a more in-depth discussion with you, and we welcome you to give us more suggestions. If you have additional suggestions, please let us know, and we will try to respond as quickly as possible.

---

> > ### Author Response · Authors · 2024-11-28
> > **Desire for Re-assessment**
> >
> > Dear Reviewer AwNJ,
> >
> > As the deadline approaches, we wanted to gently follow up on our recent submission. Your feedback is highly valuable to us, and we would greatly appreciate any updates or further guidance you may have regarding our revisions and responses.
> >
> > We sincerely hope that our responses will encourage a fresh re-evaluation of our work, and we extend our deepest gratitude for your time and consideration.

---

> > > ### Author Response · Authors · 2024-12-02
> > > **Gratitude for the Pleasant and Constructive Discussion**
> > >
> > > As the discussion period nears its conclusion, we would like to once again express our sincere gratitude for your thoughtful reviews and valuable feedback.
> > >
> > > If you are satisfied with the revisions (as detailed in the global responses) and the clarifications provided, we would greatly appreciate it if you could consider updating your review to reflect the efforts we have made to address your suggestions and improve the work.
> > >
> > > Thank you once again for your time and consideration.

---

### Author Response · Authors · 2024-11-20
**Global Response to All Reviewers (1/2)**

We thank all the reviewers for their valuable time and constructive comments. We are grateful for the opportunity to discuss and improve our paper based on your feedback.

**We have provided a revised version and highlighted the changes in blue text. In this revision, we have addressed the concerns and issues raised. We kindly request the reviewers to review our revised paper.** Below is a brief summary of the updates:

1. **Experiments on Larger LLMs:** We implemented MeCo on Llama-3-70b-Instruct and Llama-3-70b-Instruct-sft (the latter being fine-tuned with tool use queries) and validated MeCo’s effectiveness on them. The results, presented in Table 1 of the revised paper, show that MeCo improved tool use decisions on larger LLMs, particularly in without-context evaluation settings, achieving a 10.8% improvement for Llama-3-70b-Instruct and a 3.5% improvement for Llama-3-70b-Instruct-sft.
2. **Expansion of MeCa Benchmark and More Comprehensive Evaluations:** Post-submission, we continued refining the MeCa benchmark (details provided in Section 4 and Appendix A), increasing its scale, and incorporating more complex and realistic queries to better mimic real-world scenarios. We evaluated MeCo on six different tasks within MeCa-Tool and presented the results in Table 2 of the revised paper. Overall, MeCo outperformed the baselines in most evaluation settings, with only a few losses to PYes, highlighting its potential for practical deployment and effectiveness in diverse and realistic scenarios.
3. **Corrections and Improvements:** We thank the reviewers for the careful inspections and we have fixed all editing and grammatical issues in the revision, including confusing sentences, citations, typos, etc. Additionally, we have clarified the points of confusion raised by the reviewers and provided necessary explanations and contexts that were absent in the previous version.

---

### Author Response · Authors · 2024-12-02
**Global Response to All Reviewers (2/2)**

We would like to express our sincere gratitude for the valuable feedback and proactive engagement throughout the review process. We truly appreciate the recognition of our work from the reviewers, including clear problem identification and strong motivation (**AwNJ**), novel methodology (**QSLD, FudH**), extensive and strong empirical results (**AwNJ, QSLD, FudH, h45z**), and practical approach for applications (**AwNJ, QSLD**).

It has been a constructive and pleasant discussion, and we believe our manuscript has significantly improved based on your insightful suggestions. We have carefully addressed the concerns you raised and incorporated your feedback. As such, we kindly ask the reviewers to re-evaluate the revised manuscript. Below is a brief recap of the key contributions of our paper:

1. **Introduction of Adaptive Tool Use:** We introduce the novel concept of adaptive tool use, a critical yet overlooked aspect in both academia and industry. We also integrate RAG within a unified framework, addressing both latency and robustness issues in Tool Use/RAG.
2. **The MeCo Algorithm:** We propose MeCo, a lightweight, low-overhead, plug-and-play algorithm that enables LLMs to self-assess their capabilities and determine when tool use/RAG is necessary. MeCo has high compatibility with main existing high-performance inference engines. Despite its simplicity, MeCo provides significant performance improvements, even for fine-tuned models.
3. **The MeCa Benchmark:** We present MeCa, a new benchmark consisting of over 7,000 carefully validated queries to evaluate adaptive tool use and RAG awareness in LLMs. MeCa fills the gap for a comprehensive evaluation framework in this area.
4. **Extensive Experimental Validation:** We conducted extensive experiments on multiple backbone models (including a Llama-3-70B) across multiple benchmarks. The results highlight MeCo’s superior performance and its balance between effectiveness and efficiency, demonstrating its practical applicability in real-world scenarios.

---

For the reviewers' convenience in identifying the revisions and addressing their concerns, we have provided a detailed list of updates below:

1. $\textrm{\color{blue}Section 3}$ (Page 3): Added a definition of meta-cognition in the context of tool use and RAG in LLMs (Reviewer **h45z**).
2. $\textrm{\color{blue}Section 4}$ (Page 5): Enhanced the MeCa benchmark by increasing its scale and including more complex and realistic queries (more than 7000 queries) to better simulate real-world scenarios (Reviewer **AwNJ**, **QSLD**, **FudH**, **h45z**).
3. $\textrm{\color{blue}Table 1}$ (Page 8). Added empirical results on a larger backbone model, Llama-3-70B-Instruct (Reviewer **QSLD**, **FudH**, **h45z**).
4. $\textrm{\color{blue}Table 2}$ (Page 9). Expanded experimental results on the enriched MeCa benchmark, further validating MeCo’s effectiveness in more complex scenarios (Reviewer **AwNJ**, **QSLD**, **FudH**, **h45z**).
5. $\textrm{\color{blue}Section 6.1}$ (Page 10): Provided additional interpretation of the results, highlighting the superiority of MeCo (Reviewer **h45z**).
6. $\textrm{\color{blue}Appendix A}$ (Page 15): Included detailed statistics of the new MeCa benchmark (Reviewer **AwNJ**, **QSLD**, **FudH**, **h45z**).
7. $\textrm{\color{blue}Appendix C.1}$ (Page 17): Added CoT prompts and clarified why CoT is not ideal for adaptive tool use/RAG (Reviewer **h45z**).
8. $\textrm{\color{blue}Minor Updates}$: Corrected typos, updated references, added clarifications, fixed minor editing issues (Reviewer **FudH**).
9. $\textrm{\color{blue}Follow-up Updates}$ (Appendix): Will include clarification on efficiency and the CoT effect discussion in the appendix (Reviewer **h45z**).

---

If you are satisfied with the revisions, we would greatly appreciate it if you could consider updating your review to reflect the efforts we have made to improve the quality of the work based on your valuable suggestions.

Thank you once again for your time and consideration.

Warm regards,
The Authors

---

### Meta-Review · Area_Chair_v83a · 2024-12-19

**Metareview:**

The paper tackles the problem of LLM tool usage. In particular, they propose a method to have the LLM identify whether a tool is required as opposed to the LLM relying on its internal knowledge. The reviews note this task to be novel and have potential. In particular, the motivation given in the paper is mentioned to be convincing. Beyond identifying the problem and offering a solution for it, the authors provide a new dataset helping to evaluate future solutions to the problem.

The two main concerns relate to (1) the quality of the solution, (2) the quality of the writing. For (1), AwNJ mentioned the solution seems to be overly simplistic and difficult to tune, and h45z mentioned the effectiveness is limited (“While the current experimental outcomes demonstrate some level of enhancement, the improvements are relatively modest, particularly concerning larger models”). For (2) there were multiple particular issues pointed out by both FudH and h45z. The authors provide a new version, but it did not fully mitigate the concerns (see e.g. the comment by h45z about minimizing the creation of new concepts.

Both of the mentioned concerns remain after the discussion phase and are too significant to overlook. The paper appears to have potential, but in its current state it is not yet ready to be published.

**Additional Comments On Reviewer Discussion:**

see meta review

---

### Decision · Program_Chairs · 2025-01-22

Reject